# MedReason-Dx: Benchmarking Step-by-Step Reasoning of Language Models in Medical Diagnosis

## Abstract

In high-stakes domains like medicine, **how** an AI arrives at an answer can be as critical as the answer itself. However, existing medical question answering benchmarks largely ignore the reasoning process, evaluating models only on final answer accuracy. This paper addresses the overlooked importance of reasoning path evaluation in medical AI. We introduce **MedReason-Dx**, a novel benchmark that assesses not just answers but the step-by-step reasoning behind them. MedReason-Dx provides expert-annotated step-by-step solutions for both multiple-choice and open-ended questions, spanning 24 medical specialties. By requiring models to produce and be evaluated on intermediate reasoning steps, our benchmark enables rigorous testing of interpretability and logical consistency in medical QA. We present the design of MedReason-Dx and outline diverse evaluation metrics that reward faithful reasoning. Using this benchmark, we identify critical gaps in existing large language models, including domain-specific medical LLMs, particularly in their ability to handle complex diagnostic reasoning and reliably recall relevant medical knowledge. Our analysis further highlights differences between general-purpose and medical LLMs in reasoning performance. We believe this resource will advance the development of robust, interpretable medical decision support systems and foster research into large language models that can reason as well as they respond.

## 1 Introduction

Artificial intelligence systems for healthcare must not only deliver correct answers but also provide faithful reasoning. In clinical decision support and medical question answering (QA), the reasoning path leading to an answer is critical for trust and safety. A model that arrives at a diagnosis through flawed logic or guesswork poses significant risks, even if the final answer is correct. Conversely, a model that articulates its reasoning enables practitioners to verify each step, ensuring the conclusion is sound and clinically valid.

Despite this importance, most existing benchmarks for medical AI evaluate models solely on whether the final answer is right, with little or no assessment of the reasoning process. This gap is problematic in high-stakes domains: evaluating only end answers may overlook dangerous reasoning errors and fails to encourage the development of models that "think" in a human-like, transparent manner.

Recent advances in large language models (LLMs) have brought reasoning to the forefront of AI research Guo et al. (2025); Chen et al. (2025). With proper training strategies or *chain-of-thought* (CoT) prompting, LLMs can generate step-by-step solutions to complex problems, from math and logic puzzles to medical questions. By configuring models to articulate intermediate steps, researchers have achieved improved performance on challenging tasks and gained insight into model decision-making. For example, state-of-the-art medical LLMs can now produce explanations or rationales alongside their answers, showcasing the potential of AI to handle intricate clinical reasoning. These developments underscore an urgent need for benchmarks that can evaluate not just final accuracy but the quality of reasoning LLMs employ. If a model is prompted to reason but we lack ground truth reasoning paths for comparison, we cannot rigorously assess whether the model's reasoning is correct, complete, or clinically valid.

Several medical QA datasets and benchmarks have emerged, yet they predominantly focus on answer correctness. Standard benchmarks drawn from medical exams (e.g., USMLE-style question banks, MedQA (Jin et al., 2021) and MedMCQA Pal et al. (2022)) and research datasets like PubMedQA (Jin et al., 2019) have driven progress in factual recall and question answering. Some of these resources include a short explanation or reference for the answer, but they do not provide a detailed, stepwise reasoning chain that could be used to evaluate a model's thought process. In other words, existing benchmarks treat reasoning as an implicit skill, not an explicit target of evaluation. A model might earn full marks by selecting the correct option in a multiple-choice question, while in reality it could have arrived at that answer via incorrect assumptions or lucky guesswork. Conversely, a model might demonstrate mostly correct reasoning and make a minor error at the final step, but current benchmarks would simply mark the entire answer as wrong, offering no credit for nor analysis of the model's reasoning ability. This limitation hampers the development of robust medical AI: it is difficult to discern whether improvements in accuracy are due to better reasoning or just better pattern matching, and it provides no incentive for models to output interpretable solutions.

To address these challenges, we propose **MedReason-Dx**, a new benchmark explicitly designed to evaluate chain-of-thought reasoning in medical question answering. MedReason-Dx (where "Dx" denotes Diagnosis) introduces several key innovations to the evaluation of medical AI:

- **Expert-annotated reasoning chains:** Each question in MedReason-Dx is accompanied by a step-by-step solution path crafted by medical experts. These reasoning chains detail the logical steps required to arrive at the correct answer, including relevant clinical facts, intermediate inferences, and elimination of distractors in the case of multiple-choice items. This provides a gold-standard trace of correct reasoning against which model-generated solutions can be compared.

- **Diverse question formats and topics:** MedReason-Dx covers a broad spectrum of medical knowledge through both multiple-choice and open-ended questions, ensuring that models are evaluated across varied response formats. The questions span 24 medical specialties, ranging from internal medicine and cardiology to pediatrics, surgery, and more. This diversity reflects the real-world breadth of medical practice and ensures that the benchmark evaluates reasoning across different sub-domains and problem types (diagnosis, treatment decisions, biomedical mechanism explanations, etc.).

- **Evaluation metrics for reasoning quality:** MedReason-Dx advances beyond simple accuracy measures by introducing multiple metrics to evaluate the fidelity and relevance of a model's reasoning. To improve robustness across different LLM evaluators, we introduce a keypoint-based strategy: medical experts extract keypoints from ground-truth answers, and model outputs are evaluated by matching their extracted key points against these references, complementing step-wise matching. By aligning responses at the keypoint level rather than surface language, this approach reduces variability from linguistic complexity and mitigates instability across evaluators. Overall, by quantifying reasoning quality, MedReason-Dx encourages models that are not only correct, but correct for the right reasons.

- **Interpretability and robustness focus:** By requiring and evaluating intermediate reasoning, MedReason-Dx places interpretability at the core of model assessment. This is especially crucial for medical AI systems that clinicians need to trust. A model that can articulate a valid reasoning chain is inherently more transparent and easier to debug than one that only outputs an answer. Furthermore, focusing on reasoning helps reveal when a model's knowledge is superficial. We anticipate that models performing well on MedReason-Dx will demonstrate greater robustness, as they must handle complex multi-step problems in a principled way rather than relying on shallow cues. Our benchmark thus serves as a stress test for genuine reasoning ability in medical contexts.

- **Revealing gaps in existing LLMs for medical reasoning:** MedReason-Dx exposes critical limitations of current LLMs, including domain-specific medical LLMs, in handling complex diagnostic reasoning and reliably recalling relevant medical knowledge. Our evaluations also investigate differences between general-purpose and medical LLMs, providing new insights into their respective strengths and weaknesses in clinical reasoning.

In summary, MedReason-Dx is the first benchmark to comprehensively evaluate reasoning paths in medical diagnosis from multiple perspectives. It provides the community with a testbed for developing

models that aspire to be not just answer engines, but reliable reasoning assistants in healthcare. By emphasizing how answers are derived, our work addresses a critical gap for high-stakes AI: the need for systems whose decisions can be inspected, trusted, and used responsibly in clinical practice.

## 2 RELATED WORKS

### 2.1 MEDICAL LLMs

Medical large language models (Med-LLMs) have advanced rapidly through innovations in model architectures, training paradigms, and domain-specific adaptations, enabling applications in information extraction, clinical decision support, dialogue systems, and multimodal medical AI.

Early Med-LLMs such as BioBERT (Lee et al., 2020) and PubMedBERT (Gu et al., 2021) leveraged large-scale biomedical corpora for pretraining, achieving strong performance in named entity recognition and text classification. Subsequent models expanded capabilities to clinical text generation and summarization, including ClinicalT5 (Lu et al., 2022), GatorTron (Yang et al., 2022), and Codex-Med (Liévin et al., 2024), while Galactica (Taylor et al., 2022) targeted medical literature analysis.

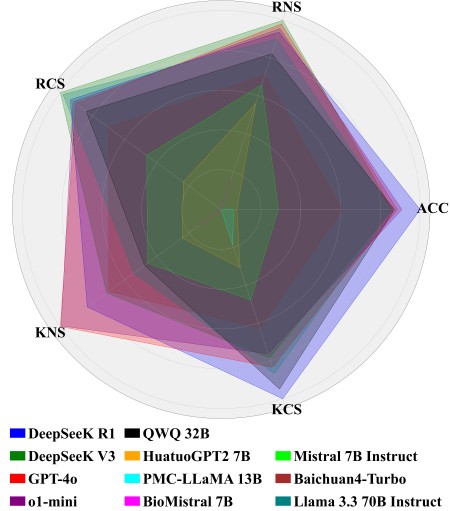

More recent systems adopt instruction fine-tuning and reinforcement learning from human feedback (RLHF) to improve reliability in clinical reasoning and QA. Med-PaLM (Singhal et al., 2023), Med-PaLM 2 (Singhal et al., 2025), and Med-Alpaca (Han et al., 2023) exemplify this trend, while GatorTronGPT (Peng et al., 2023) adapts GatorTron for precise report generation and ChatDoctor (Li

Figure 1: Medical reasoning performance of advanced LLMs in MedReason-Dx. Our benchmark evaluates LLMs' medical reasoning capabilities across five dimensions: ACC, RNS, RCS, KNS and KCS. Results indicate that medical LLMs don't outperform general LLMs on complex medical reasoning tasks.

et al., 2023) focuses on virtual consultations. Further domain specialization has produced models such as PMC-LLaMA (Wu et al., 2023a) for biomedical literature, GPT-4-Med (Nori et al., 2023) for advanced clinical tasks, and Taiyi-LLM (Luo et al., 2024) and Zhongjing (Yang et al., 2024b) for Traditional Chinese Medicine. Multilingual and multimodal efforts—such as HuatuoGPT (Zhang et al., 2023), Med-Flamingo (Moor et al., 2023), and Med-Gemini (Saab et al., 2024)—extend the scope of Med-LLMs to global and cross-modal healthcare applications.

These developments underscore the rapid evolution of Med-LLMs, enhancing their ability to process complex medical language, integrate multimodal data, and support diverse clinical tasks. As these models become more powerful and widely deployed, it is increasingly important to evaluate not only their final predictions but also the reasoning processes behind them. Without explicit reasoning-path evaluation, it remains unclear whether models reach conclusions through sound clinical logic or superficial correlations. Thus, rigorous assessment of reasoning fidelity is essential for ensuring safety, trust, and real-world reliability in medical AI.

### 2.2 MEDICAL BENCHMARKS

The development of standardized datasets and robust evaluation platforms is crucial for advancing AI in medicine. Existing efforts can be broadly grouped into two directions: (1) datasets tailored for specific medical AI tasks, and (2) automated benchmarks for assessing the capabilities of LLMs.

The first category includes datasets supporting tasks such as information extraction, QA, and natural language inference. Widely used corpora like CADEC (Karimi et al., 2015) and BC5CDR (Li et al., 2016) target biomedical NER and relation extraction. QA datasets such as MedQA (Jin et al., 2021) and PubMedQA (Jin et al., 2019) emphasize clinical knowledge retrieval and reasoning. Large-scale resources like MIMIC-III (Johnson et al., 2016), HealthSearchQA (Singhal et al., 2023), and

Table 1: Comparison with existing Medical QA benchmarks.

| Benchmark | CoT Evaluation | No. Domains | reasoning intensive | MCQ | OEQ | Expert Annotation |
|---|---|---|---|---|---|---|
| MMedBench | ✗ | 21 | ✓ | ✓ | ✗ | ✓ |
| MedQA | ✗ | - | ✗ | ✓ | ✗ | ✗ |
| MedMCQA | ✗ | 21 | ✗ | ✓ | ✗ | ✓ |
| MMLU | ✗ | 6 | ✗ | ✓ | ✗ | ✗ |
| Medbullets | ✗ | - | ✓ | ✓ | ✗ | ✓ |
| JAMA Challenge | ✗ | 13 | ✓ | ✓ | ✗ | ✓ |
| LiveQA | ✗ | - | ✗ | ✗ | ✓ | ✓ |
| ClinicBench | ✗ | - | ✗ | ✓ | ✓ | ✓ |
| **Ours** | ✓ | 24 | ✓ | ✓ | ✓ | ✓ |

CORD-19 (Wang et al., 2020) support report summarization, while MedNLI (Romanov & Shivade, 2018) focuses on natural language inference. Recently, MedReason (Wu et al., 2025a) addresses the lack of high-quality reasoning data by constructing 32,682 QA pairs with detailed, knowledge–graph–guided Chain-of-Thought explanations, further validated by expert review. Similar efforts such as MedCaseReasoning (Wu et al., 2025b) also highlights reasoning evaluation in clinical cases.

Table 2: Model Comparison.

| Models | Params |
|---|---|
| *General Large Language Models* | |
| GPT-4o (OpenAI, 2024a) | - |
| GPT-o1 mini (OpenAI, 2024b) | - |
| DeepSeek R1 (Guo et al., 2025) | 671B |
| DeepSeek V3 (Liu et al., 2024) | 671B |
| Mistral (Jiang et al., 2023) | 7B |
| LLAMA 3.3 (Dubey et al., 2024) | 70B |
| QWQ (Yang et al., 2024a) | 32B |
| *Medical Large Language Models* | |
| HuatuoGPT2 (Chen et al., 2023) | 7B |
| BioMistral (Labrak et al., 2024) | 7B |
| PMC-LlaMA (Wu et al., 2024) | 13B |
| Baichuan4-Turbo (Yang et al., 2023) | - |

Table 3: Statistics of MedReason-Dx.

| Statistic | Number |
|---|---|
| Total questions number | 1170 |
|   - Multiple-choice questions | 592 |
|   - Open-ended questions | 578 |
| Maximum question length | 578 |
| Maximum answer length | 1028 |
| Average question length | 166.9 |
| Average answer length | 320.0 |
| Maximum number of steps | 13 |
| Maximum number of key points | 118 |
| Minimum number of steps | 3 |
| Minimum number of key points | 3 |
| Average number of steps | 6.4 |
| Average number of key points | 27.1 |

The second category emphasizes automated evaluation frameworks that reduce reliance on costly expert review. MedBench (Cai et al., 2024) provides a broad platform with 40,041 questions spanning multiple fields. AutoEval (Liao et al., 2023) reformulates USMLE problems into multi-turn dialogues for assessing information coverage and task accuracy. LLM-Mini-CEX (Shi et al., 2023) employs patient simulators and ChatGPT to evaluate diagnostic dialogue quality, while MedGPTEval (Xu et al., 2023) integrates Chinese datasets and public benchmarks with 16 expert-curated indicators. LLM-Human Evaluation (Chiang & Lee, 2023) further explores the feasibility of automated assessment, demonstrating alignment with human experts on adversarial and open-ended tasks. Collectively, these resources provide systematic, low-cost ways to benchmark AI models and support optimization.

## 3 MEDICAL BENCHMARK WITH STEP-WISE EVALUATION

### 3.1 DATA CURATION

#### 3.1.1 DATA CURATION FOR MULTIPLE-CHOICE QUESTION

The data collection for MedReason-Dx is designed to construct a challenging reasoning dataset that goes beyond typical knowledge-recall question answering. Our goal is to curate problems that require complex, multi-step reasoning, reflecting the intricate workflows of real-world clinical diagnostics. To this end, we employ a rigorous data selection strategy, filtering questions from well-established medical datasets that encompass authentic clinical cases across multiple disciplines.

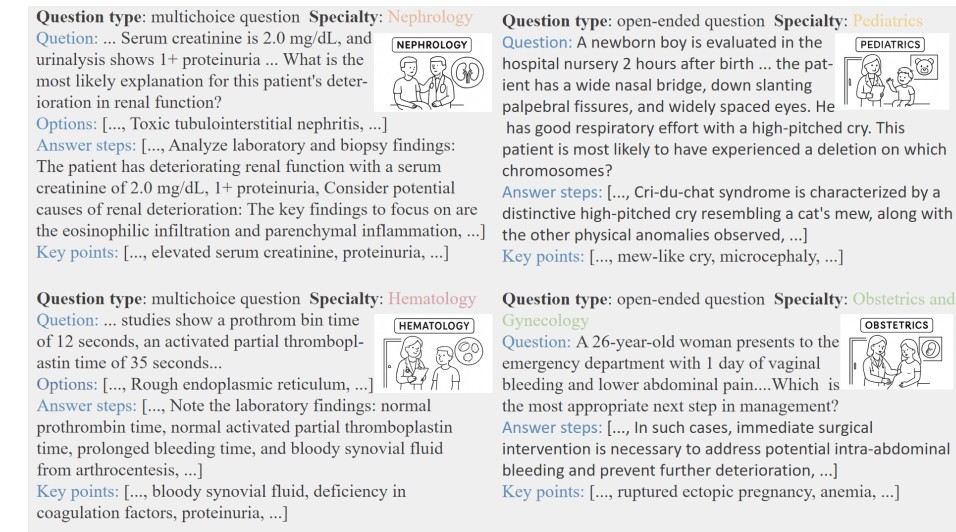

Figure 2: Dataset overview of MedReason-DX.

Table 4: (COT evaluation) Benchmarking the medical reasoning performance of existing models

|  | multiple-choice | | | | open-ended | | | |
|---|---|---|---|---|---|---|---|---|
|  | RNS | RCS | KNS | KCS | RNS | RCS | KNS | KCS |
| DeepSeek R1 | 86.11 | 73.46 | 47.62 | 33.90 | 93.04 | 60.97 | 40.68 | 31.20 |
| DeepSeek V3 | 91.09 | 73.76 | 44.49 | 26.06 | 94.78 | 67.52 | 39.54 | 32.08 |
| GPT-4o | 89.92 | 69.76 | 50.72 | 29.78 | 93.88 | 60.97 | 43.53 | 29.93 |
| o1-mini | 89.94 | 69.15 | 51.11 | 29.46 | 91.22 | 63.30 | 42.97 | 27.95 |
| Mistral 7B Instruct | 72.37 | 38.26 | 40.21 | 24.65 | 76.85 | 43.48 | 34.78 | 23.60 |
| Llama 3.3 70B Instruct | 86.98 | 72.71 | 41.55 | 31.54 | 88.32 | 66.75 | 36.47 | 29.26 |
| QWQ 32B | 84.18 | 65.73 | 40.31 | 33.92 | 82.65 | 57.72 | 35.33 | 29.51 |
| Baichuan4-Turbo | 64.49 | 56.47 | 45.40 | 26.39 | 90.32 | 51.75 | 38.13 | 26.64 |
| HuatuoGPT2 7B | 62.59 | 24.13 | 35.59 | 20.57 | 76.14 | 32.36 | 31.74 | 22.21 |
| PMC-LLaMA 13B | 39.91 | 17.95 | 30.22 | 20.36 | 38.77 | 12.78 | 28.49 | 18.51 |
| BioMistral 7B | 44.72 | 13.97 | 31.58 | 15.60 | 50.83 | 16.43 | 31.96 | 17.23 |

To capture the breadth of medicine, we define 24 domains aligned with common hospital departments, including "Cardiology", "Pulmonology", "Gastroenterology" and so on, as shown in Figure 4. The selection of questions prioritizes diversity in the types of clinical challenges and the reasoning methods required for problem-solving. This diversity encompasses a wide array of diagnostic tasks that span both common and rare clinical conditions. Questions are deliberately selected for their requirement of complex multi-step reasoning, including, but not limited to, physiological mechanism analysis, differential diagnosis, hypothesis testing, exclusionary reasoning, and the integration of cross-disciplinary knowledge, while simple factual recall items (e.g., "What is the normal body temperature?") are excluded. By focusing on the reasoning complexity and diversity, MedReason-Dx reflects the multifaceted nature of clinical decision-making and the diverse set of cognitive strategies employed by healthcare professionals in practice. The aim is to ensure that the dataset not only captures the breadth of medical knowledge but also challenges models to engage in higher-order reasoning reflective of real-world medical diagnostic scenarios

Following the selection of complex questions, human experts from diverse specialties annotate step-by-step solutions and key points, with LLMs providing supplementary support. This methodology enables a thorough assessment of the model's reasoning capabilities, evaluating not only the accuracy of the final answer but also the clarity and logical coherence of the reasoning process. The step-by-step solutions deconstruct the reasoning into concise, logical steps, each reflecting a critical component of the decision-making process. Key points emphasize essential information, such as clinical findings or diagnostic considerations, required for accurate diagnosis. These annotations ensure that the model's

response addresses all relevant aspects of medical decision-making. The objective is to rigorously evaluate the reasoning process, confirming its completeness and logical integrity.

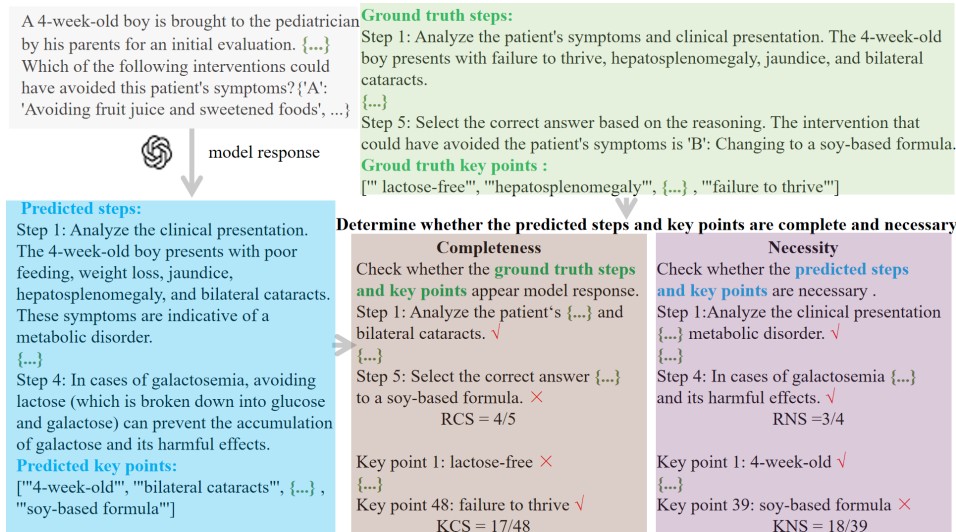

Figure 3: Illustration of the proposed evaluation metrics. For LLM responses, we match them with annotations from human experts to evaluate the necessity and sufficiency of the reasoning process.

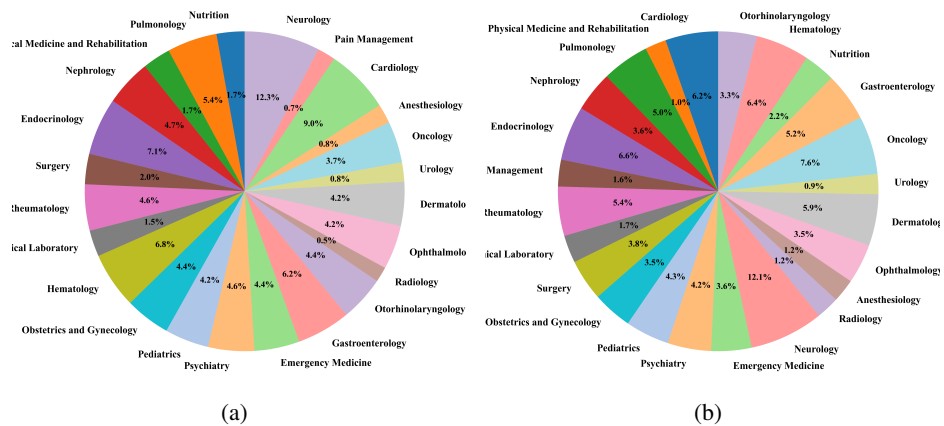

Figure 4: Medical Specialty Distribution of MedReason-Dx, including (a): multiple-choice question and (b) open-ended question, both covering 24 departments.

### 3.1.2 DATA CURATION FOR OPEN-ENDED QUESTION

Multiple-choice questions often fail to reflect real-world clinical scenarios, where physicians make decisions without predefined options. To address this limitation, we develop additional open-ended reasoning questions. To ensure diversity, the dataset incorporates both multiple-choice and open-ended sources. Multiple-choice questions are adapted into open-ended formats by using LLMs to revise only the final question sentence, preserving the clinical context while minimizing alterations. All modifications are reviewed by human experts to ensure accuracy. Once open-ended questions are established, domain specialists generate step-by-step solutions using the same methodology as for multiple-choice items. And key points are then extracted to enable systematic evaluation.

In summary, MedReason-Dx provides a comprehensive benchmark for evaluating LLMs on complex medical reasoning tasks through carefully selected questions and detailed expert annotations. Detailed statistics for MedReason-Dx are presented in Table 3 and comparisons with other relevant medical benchmarks are provided in Table 1. More details are provided in Appendix A.4.

## 3.2 EVALUATION

To rigorously evaluate the reasoning ability of LLMs, we propose a comprehensive framework that assesses three critical dimensions: the accuracy of the final answer, the quality of the reasoning process, and the incorporation of relevant key points. These dimensions are captured through five metrics: *Accuracy*, *Reasoning Completeness Score (RCS)*, *Reasoning Necessity Score (RNS)*, *Key point Completeness Score (KCS)*, and *Key point Necessity Score (KNS)*. Below, we detail each metric, its motivation, and the computational methodology, with a particular emphasis on the thoroughness and necessity of reasoning steps and key points usage.

### 3.2.1 ACCURACY

The *Accuracy* metric evaluates the correctness of the model's final answer, a standard measure of performance in question-answering tasks. For multiple-choice questions, we compare the model's selected option with the ground-truth answer. For open-ended questions, the model is instructed to provide answers in a standardized format: *Therefore, the answer is {your answer}*. And a LLM is employed to assess semantic equivalence between the model's answer and the ground-truth answer, accommodating variations in expression.

The Accuracy is computed as:

$$\text{Accuracy} = \frac{\sum_{i=1}^{N} \mathbb{I}(\hat{y}_i = y_i)}{N}, \quad (1)$$

Table 5: Benchmarking the accuracy (%) performance of existing models.

| | Multiple-choice | | Open-ended | |
|---|---|---|---|---|
| | CoT | Direct | CoT | Direct |
| DeepSeek R1 | 65.03 | 64.36 | 40.14 | 42.39 |
| DeepSeek V3 | 60.47 | 59.79 | 33.56 | 37.02 |
| GPT-4o | 58.28 | 59.12 | 37.72 | 47.70 |
| o1-mini | 60.47 | 62.67 | 37.37 | 41.18 |
| Mistral 7B Instruct | 30.74 | 18.58 | 17.47 | 15.57 |
| Llama 3.3 70B Instruct | 56.42 | 57.60 | 37.02 | 37.89 |
| QWQ 32B | 56.08 | 55.91 | 38.75 | 40.83 |
| Baichuan4-Turbo | 46.62 | 42.74 | 27.16 | 28.37 |
| HuatuoGPT2 7B | 19.26 | 25.00 | 12.28 | 12.81 |
| PMC-LLaMA 13B | 20.95 | 23.14 | 9.17 | 6.23 |
| BioMistral 7B | 15.71 | 19.76 | 9.52 | 12.63 |

where $N$ is the total number of questions, $\hat{y}_i$ is the model's predicted answer for the $i$-th question, $y_i$ is the ground-truth answer, and $\mathbb{I}(\cdot)$ is the indicator function, returning 1 if the predicted answer matches the ground truth and 0 otherwise. For open-ended questions, equivalence is determined by the LLM, ensuring robust evaluation across diverse answer formats.

### 3.2.2 REASONING COMPLETENESS SCORE (RCS)

The *Reasoning Completeness Score (RCS)* quantifies the extent to which the model's reasoning includes all critical steps required to derive the answer. This is crucial in high-stakes domains, where comprehensive reasoning ensures all relevant concepts and logical deductions are addressed.

For each question, a reference set of essential reasoning steps $R_i = \{r_1, r_2, \ldots, r_s\}$ is predefined based on expert annotations. The model's reasoning text is evaluated to identify the presence of these steps or their semantic equivalents. The RCS for a single question is:

$$\text{RCS}_i = \frac{|\hat{R}_i \cap R_i|}{|R_i|}, \quad (2)$$

where $\hat{R}_i$ is the set of reasoning steps in the model's response for the $i$-th question, and $|\cdot|$ denotes set cardinality. The overall RCS is:

$$\text{RCS} = \frac{1}{N} \sum_{i=1}^{N} \text{RCS}_i. \quad (3)$$

### 3.2.3 REASONING NECESSITY SCORE (RNS)

The *Reasoning Necessity Score (RNS)* evaluates the conciseness of the reasoning process, penalizing extraneous or redundant steps. This ensures that the model's reasoning is focused and interpretable, a critical requirement for practical utility in medical contexts.

RNS is computed by comparing the model's responses $\hat{R}_i$ to the reference set $R_i$. Steps in $\hat{R}_i$ that are not in $R_i$ or its semantic equivalents are considered unnecessary. RNS is calculated as follows:

$$\text{RNS} = \frac{1}{N} \sum_{i=1}^{N} 1 - \frac{|\hat{R}_i \setminus R_i|}{|\hat{R}_i| + \epsilon}, \tag{4}$$

where $\epsilon > 0$ (e.g., $\epsilon = 0.01$) prevents division by zero.

### 3.2.4 KEY POINT COMPLETENESS SCORE (KCS)

The *Key point Completeness Score (KCS)* assesses the model's ability to incorporate all relevant domain-specific key points in its reasoning, such as diagnoses, symptoms, or treatments in medical contexts. Completeness is critical to ensure that the model captures essential domain knowledge.

For each question (i), a reference set of key points $K_i = \{k_1, k_2, \ldots, k_w\}$ is annotated by human experts, representing the essential terms relevant to the correct response. The model's response is analyzed to extract a set of key points $\hat{K}_i = \{\hat{k}_1, \hat{k}_2, \ldots, \hat{k}_{\hat{w}}\}$, which denotes the terms provided by AI model. KCS is calculated as follows:

$$\text{KCS} = \frac{1}{N} \sum_{i=1}^{N} \frac{|\hat{K}_i \cap K_i|}{|K_i|}. \tag{5}$$

### 3.2.5 KEY POINTS NECESSITY SCORE (KNS)

The *Key points Necessity Score (KNS)* evaluates the model's ability to avoid irrelevant or extraneous key points, ensuring that the reasoning remains focused and relevant.

The KNS is computed by identifying key points in $\hat{K}_i$ that are not in the reference set $K_i$:

$$\text{KNS} = \frac{1}{N} \sum_{i=1}^{N} 1 - \frac{|\hat{K}_i \setminus K_i|}{|\hat{K}_i| + \epsilon}. \tag{6}$$

The proposed evaluation framework, comprising Accuracy, RCS, RNS, KCS, and KNS, extends model assessment beyond final answer correctness to capture reasoning quality and knowledge relevance. Specifically, RCS measures the completeness of reasoning steps, RNS evaluates conciseness, KCS verifies coverage of critical medical terms, and KNS assesses the relevance of recalled knowledge within the reasoning process. Together, these metrics provide a comprehensive evaluation of both correctness and interpretability, which is essential for trustworthy deployment in high-stakes medical applications. An overview of the metric calculations is shown in Figure 3.

## 4 EXPERIMENTS

### 4.1 EXPERIMENTAL SETUP

**Evaluation Models.** To provide a comprehensive benchmark, we conduct evaluations on 11 advanced LLMs, comprising 7 general LLMs and 4 medical LLMs. For general LLMs, we include DeepSeek R1 (Guo et al., 2025), DeepSeek V3 (Liu et al., 2024), GPT-4o (OpenAI, 2024a), o1-mini (OpenAI, 2024b), Mistral 7B Instruct (Jiang et al., 2023), LlaMA3.3 70B Instruct (Dubey et al., 2024), and QWQ 32B (Yang et al., 2024a). For medical LLMs, we include Baichuan4-Turbo (Yang et al., 2023), HuatuoGPT2-7B (Chen et al., 2023), PMC-LlaMA 13B (Wu et al., 2024) and BioMistral 7B (Labrak et al., 2024). A comparison of these models is provided in Table 2.

**Implementation Details.** Following previous work (Jiang et al., 2025), we leverage two types of prompts to guide the model to give answers: chain-of-thought (CoT) prompts (Wei et al., 2022) and direct prompts. When calculating RNS, RCS, KNS, and KCS scores, we utilize the model's CoT response to explicitly require the model to provide step-by-step answers. When calling LLMs, the temperature is set to 0.7, Top-P is set to 0.9, and max tokens is set to 1024. In the experiment, we employ gpt4o-mini (Achiam et al., 2023) as the judge. We provide detailed prompts in Appendix A.6.

## 4.2 BENCHMARKING MEDICAL LLMS

**LLMs struggle with complex medical reasoning tasks.** As shown in Table 5, we benchmark several advanced language models on the MedReason-Dx dataset under two prompting settings: Chain-of-Thought (CoT) and Direct Answering, across both multiple-choice and open-ended formats. Our findings reveal that state-of-the-art large language models, including GPT-4o and DeepSeek-R1, continue to encounter substantial difficulties in addressing complex medical reasoning tasks. Specifically, in multiple-choice question assessments, the highest-performing model, DeepSeek-R1, achieved an accuracy of 65.03%. In contrast, performance on open-ended questions was notably lower, with the leading model attaining only 47.70% accuracy (GPT-4o). These suboptimal results underscore the persistent limitations of current large language models in navigating the intricacies of real-world medical reasoning scenarios, highlighting the critical need for further advancements and optimization in model development.

**LLMs face difficulties in comprehensively recalling medical knowledge.** As shown in Table 4, to thoroughly evaluate the reasoning capabilities of LLMs beyond the correctness of their final answers, we propose four novel metrics: RNS, RCS, KNS and KCS. These metrics enable a comprehensive assessment of the model's step-by-step reasoning process and the

Table 6: Evaluation using different judges.

| Judge | GPT-4o | | DeepSeeK R1 | |
|---|---|---|---|---|
| | RNS | RCS | RNS | RCS |
| GPT-4o-mini | 89.92 | 69.76 | 86.11 | 73.46 |
| GPT-4.1-mini | 88.47 | 72.63 | 88.08 | 73.83 |

relevance of key points used during reasoning. Our results indicate that each reasoning step, along with the key points mentioned in the reasoning process, is typically critical for accurate problem-solving. However, LLMs often struggle to include all the necessary information annotated by human experts. For instance, GPT-4o achieved an RNS of $89.92\%$ on multiple-choice questions, reflecting high precision in its reasoning process with minimal inclusion of irrelevant details. In contrast, its RCS of $69.76\%$ highlights a significant challenge in recalling all the essential information required to solve problems as effectively as human experts. This gap is likely attributable to the complexity and irregularity of medical knowledge, which lacks the structured theorems and corollaries characteristic of fields like mathematics and physics, posing unique challenges for solving medical problems.

**Medical LLMs don't outperform general LLMs in complex medical reasoning tasks.** As illustrated in Table 4 and 5, medical LLMs don't surpass general LLMs. This observed disparity may stem from prevailing training paradigms for medical LLMs, which primarily entail the assimilation of foundational medical corpora while largely omitting the integration of sophisticated datasets designed to cultivate advanced medical reasoning competencies. Consequently, although medical LLMs may demonstrate marginal advantages over general LLMs in rudimentary medical knowledge retrieval and question-answering tasks, their proficiency in multifaceted, real-world clinical contexts frequently proves inadequate relative to anticipated benchmarks. This observation highlights the critical need for developing comprehensive training datasets specifically tailored to large-scale, complex medical reasoning tasks to comprehensively enhance the capabilities of medical LLMs.

## 5 EVALUATION CONSISTENCY ACROSS DIFFERENT LLM JUDGES

In our study, we employ GPT-4o-mini as the judge to assess the similarity between model-generated responses and the ground truth. To validate the robustness of our results, we conduct additional experiments using GPT-4.1-mini as the judge and compare the consistency across both evaluation sets. The experiments are performed on responses generated by GPT-4o and DeepSeek-R1. As reported in Table 6, the scores derived from the two independent judges exhibit a high degree of agreement, substantiating the reliability and validity of our evaluation framework.

## 6 CONCLUSION

In this paper, we introduce MedReason-Dx, a novel benchmark designed to evaluate not only the accuracy of medical question answering systems but also the quality of their reasoning processes. By providing expert-annotated, step-by-step reasoning chains and evaluating models across multiple medical specialties, MedReason-Dx promotes the development of AI systems that can articulate their reasoning, ensuring both interpretability and robustness. Our approach highlights the critical importance of evaluating the reasoning process behind AI-generated answers, fostering the creation of transparent, reliable, and clinically valid decision support tools in healthcare.

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

## A APPENDIX

### A.1 LLM USAGE STATEMENT

In the preparation of this manuscript, we utilized Large Language Models (LLMs) to assist with language polishing and refinement of the text. Specifically, the LLM was employed to enhance the clarity, coherence, and grammatical accuracy of the writing, ensuring that the manuscript adheres to high standards of academic communication. The LLM did not contribute to the research ideation, methodology, data analysis, or core content development, which were entirely conducted by the authors. All outputs generated by the LLM were carefully reviewed and edited by the authors to ensure alignment with the intended scientific contributions and to maintain the integrity of the work.

### A.2 OPEN-ENDED QUESTIONS POSE A SIGNIFICANTLY GREATER CHALLENGE TO LLMS THAN MULTIPLE-CHOICE QUESTIONS.

In clinical practice, physicians frequently address open-ended questions reflective of real-world medical scenarios. However, many existing medical evaluation benchmarks predominantly utilize multiple-choice formats, which substantially lower the complexity of questions and deviate from authentic clinical contexts. This discrepancy may result in overly optimistic evaluations of LLMs capabilities. As evidenced in Tables 5 and 4, LLMs exhibit significantly reduced performance on open-ended questions compared to multiple-choice questions. To elucidate the differences in model responses across these question types, we administer identical questions in both formats to GPT-4o and analyzed the variations in their responses. As shown in Table 7, alterations solely in the question's format lead to substantial variations in the model's response.

### A.3 COMPARISON OF MODEL PERFORMANCE UNDER DIFFERENT PROMPTS

As show in Table 5, we benchmarked several advanced language models on the MedReason-Dx dataset under two prompting settings: Chain-of-Thought (CoT) and Direct Answering, across both multiple-choice and open-ended formats. Overall, DeepSeek-R1 achieved the highest performance in the multiple-choice setting, with CoT prompting slightly outperforming direct answering (65.03% vs. 64.36%). However, in the open-ended setting, its performance reversed, with direct prompting yielding higher accuracy (42.39%) than CoT (40.14%). DeepSeek-V3 showed a similar trend with modest gains from direct answering in open-ended questions (37.02% vs. 33.56%). Interestingly, GPT-4o exhibited the largest gap in favor of direct prompting for open-ended questions (47.70% vs. 37.72%), while maintaining comparable results in multiple-choice settings. o1-mini demonstrated relatively balanced performance across settings, with a slight edge for direct prompting in both question types. In contrast, Baichuan4-Turbo underperformed across all configurations, with particularly low scores on open-ended questions, indicating a significant gap in step-by-step reasoning capabilities compared to stronger models. These results suggest that while CoT prompting can provide marginal gains in structured formats, direct answering may be more effective in complex open-ended clinical scenarios, particularly for stronger LLMs.

## A.4 DATA COLLECTION

To ensure data diversity, MedReason-Dx compiles data from a wide range of sources, including JAMA Clinical Challenge (Chen et al., 2024), Medbullets (Chen et al., 2024), MMedBench (Qiu et al., 2024), nephSAP (Wu et al., 2023b), LiveQA (Abacha et al., 2017) and PubMedQA (Jin et al., 2019). To confirm that the selected questions demand rigorous clinical reasoning, we employed two stringent criteria for data curation: (1) questions that advanced large language models (e.g., GPT-4o, DeepSeek R1) failed to answer correctly were designated as reasoning-intensive and retained; (2) questions necessitating more extensive and detailed responses were similarly classified as reasoning-intensive and incorporated. These criteria effectively excluded questions predicated primarily on rote recall of medical knowledge. The preliminarily screened questions subsequently underwent rigorous secondary review and expert annotation by domain specialists, culminating in the development of MedReason-Dx—a robust benchmark designed to evaluate complex medical reasoning capabilities. The distribution of questions across the various sources is detailed in Table 8.

## A.5 COMPARISON OF MODEL RESPONSES

In Table 9, we present a complete data from our benchmark. In Table 10 11 12 13 we compare the differences between responses from several models. Overall, the larger the number of parameters in a model, the better its performance in generating answers.

## A.6 DETAILS OF PROMPTS

The prompt for calculating RNS is as follows:

Your task is to evaluate the correctness of each step of a model's solution.
## Input Format
1. Problem: The original question/task. Answer options are provided if available.
2. Answer: The ground truth final answer
3. Ground Truth Steps: Essential steps required for deriving the correct answer
4. A solution of a model (split into predicted steps)
## Task
For each predicted step in the model solution,
- it must exactly match or be directly entailed by the ground truth.
- if there is no direct match, the step must not contradict the ground truth and must have valid logical reasoning.
### Judgement Categories
- Match: Aligns with ground truth
- Reasonable: Valid but not in ground truth
- Wrong: Invalid or contradictory
- N/A: For background information steps
### Output Requirements
1. The output must be JSON only without any other content or formatting. Do not add "'json, etc.
2. The length of the JSON list must be the same as the number of steps in model solution.
### Output Format
[
{{
"step_{index}": <integer>, "premises": <evidence (if available)>,
"conclusion": <step conclusion>, "judgment": "Match" | "Reasonable" | "Wrong" | "N/A"
}},
...
]
## Inputs
Here are the problem, answer, model solution (predicted steps), and ground truth steps:
[Question]
{question}
{answer_options}
[Ground Truth Answer]
{answer}
[Ground Truth Steps]
{answer_steps}
[Model Solution (Predicted Steps)]
{predicted_steps}

The prompt for calculating RCS is as follows:

Your task is to match the ground truth steps with the provided model solution.
## Input Format
1. Problem: The original question/task. Answer options are provided if available.
2. Answer: The ground truth final answer
3. Ground Truth Steps: Essential steps required for deriving the correct answer
4. A solution of a model
## Matching Process
- You need to match each ground truth step with the model solution provided.
- Match Criteria:
- Each ground truth step should exactly match a part of the solution or is directly entailed by a part of the solution
- All the details in a step must be matched, including the specific value and content
- You should judge all the ground truth steps for whether there is a match in the solution
## Output Format
[
{{
"step_index": <integer>,
"judgment": "Matched" | "Unmatched"
}},
...
]
## Additional Rules
1. Only output the JSON array with no additional information.
2. Judge each ground truth step in order without omitting any step.
## Inputs
Here are the problem, answer, model solution, and ground truth steps:
[Question]
{question}
{answer_options}
[Ground Truth Answer]
{answer}
[Ground Truth Steps]
{answer_steps}
[Model Solution]
{predicted_steps}

The prompt for calculating KNS is as follows:

Your task is to match the ground truth key points with the key points provided by an AI model.
## Input Format
1. Problem: The original question/task. Answer options are provided if available.
2. Answer: The ground truth final answer.
3. Ground Truth Steps: Essential steps required for deriving the correct answer.
4. Ground Truth Key Ponits: Key points in the ground truth steps.
5. Key points provided by an AI model.
## Matching Process
- You need to determine how many of the key points provided by the AI model are correct.
- Match Criteria:
- When a given key point matches a ground truth keypoint, it means that they have similar meanings in the context of this issue.
- You should judge all the key points provided by the AI model for whether there is a match in the ground truth key points.
## Output Format
[
{{
"key_point_index": <integer>,
"key_point": <key point>,
"judgment": "Matched" | "Unmatched"
}},
... ]
## Additional Rules
1. Only output the JSON array with no additional information.
2. Judge each key point provided by the AI model in order without omitting any key point.
## Inputs
Here are the problem, answer, ground truth steps, ground truth key points and key points provided by the AI model:
[Question]
{question}
{answer_options}
[Ground Truth Answer]
{answer}
[Ground Truth Steps]
{answer_steps}
[Ground Truth Key Ponits]
{key_words}
[Key points provided by the AI model]
{provided_key_words}

The prompt for calculating KCS is as follows:

Your task is to match the ground truth key points with the key points provided by an AI model.
## Input Format
1. Problem: The original question/task. Answer options are provided if available.
2. Answer: The ground truth final answer.
3. Ground Truth Steps: Essential steps required for deriving the correct answer.
4. Ground Truth Key Points: Key points in the ground truth steps.
5. Key points provided by an AI model.
## Matching Process
- You need to determine how many of the key points provided by the AI model are actually present in the ground truth key points.
- You should pay attention to how many key points are successfully found by the AI model.
- Match Criteria:
- When a given key point matches a ground truth keypoint, it means that they have similar meanings in the context of this issue.
- You should judge all the ground truth key points for whether there is a match in the key points provided by the AI model.
## Output Format
[
{{
"key_point_index": <integer>,
"key_point": <key point>,
"judgment": "Matched" | "Unmatched"
}},
... ]
## Additional Rules
1. Only output the JSON array with no additional information.
2. Judge each ground truth key point in order without omitting any key point.
## Inputs
Here are the problem, answer, ground truth steps, ground truth key points and key points provided by the AI model:
[Question]
{question}
{answer_options}
[Ground Truth Answer]
{answer}
[Ground Truth Steps]
{answer_steps}
[Ground Truth Key Ponits]
{key_words}
[Key points provided by the AI model]
{provided_key_words}

The prompt for get CoT response is as follows:

You are a medical professional. Please answer the following questions.
[Question]
{question}
Give your answer in the following form with clear logic:
Step1: Step2:.... . Therefore, the answer is \box{{}}.

The prompt for get direct response is as follows:

You are a medical professional. Please answer the following questions.
[Question]
{question}
End your answer in this format:
Therefore, the answer is \box{}.

The prompt for extracting key points from responses is as follows:

You are a medical professional. We want to find out how many key points the AI model got right in its answer, so please extract the key points in the answer provided by the AI model based on the question and the answer. Please use as many words from the answer as possible. The oringal question is:
[question]
And the answer provided by the AI model is:
[answer]
Please put the found key points into a list: [key point 1, key point 2,...]

Table 7: Comparison of model responses to multiple-choice question and open-ended question

| | multiple-choice question | open-ended question |
|---|---|---|
| **Question** | A 31-year-old woman presents to her obstetrician for her first prenatal visit after having a positive home pregnancy test 1 week ago. Her last menstrual period was 8 weeks ago. The patient has a medical history of type 1 diabetes mellitus since childhood and her home medications include insulin. Her hemoglobin A1c 2 weeks ago was 13.7%. {...} Her temperature is 98.6°F (37.0°C), blood pressure is 124/81 mmHg, pulse is 75/min, and respirations are 14/min. A physical exam is unremarkable. This fetus is most likely at increased risk for which of the following complications? ["Aneuploidy", "Neonatal hyperglycemia", "Neural tube defect", "Oligohydramnios", "Post-term delivery"] | A 31-year-old woman presents to her obstetrician for her first prenatal visit after having a positive home pregnancy test 1 week ago. Her last menstrual period was 8 weeks ago. The patient has a medical history of type 1 diabetes mellitus since childhood and her home medications include insulin. Her hemoglobin A1c 2 weeks ago was 13.7%. ... Her temperature is 98.6°F (37.0°C), blood pressure is 124/81 mmHg, pulse is 75/min, and respirations are 14/min. A physical exam is unremarkable. This fetus is most likely at increased risk for which complications? |
| **Response** | ["Identify the key medical condition of the patient, which is type 1 diabetes mellitus with poor glycemic control, indicated by a hemoglobin A1c of 13.7%.", "Recognize that poor glycemic control in early pregnancy, especially in the periconceptional period, increases the risk of congenital malformations in the fetus.", "Understand that the most common anomalies in infants of diabetic mothers (IDM) with poor glycemic control include cardiovascular defects and neural tube defects, such as anencephaly and spina bifida.", "Review the options provided to determine which complication is most likely associated with poorly controlled diabetes in pregnancy:\n- Option A: Aneuploidy is related to advanced maternal age, not diabetes.\n- Option B: Neonatal hyperglycemia is incorrect as IDM typically experience hypoglycemia due to fetal hyperinsulinemia.\n- Option C: Neural tube defect is a known risk in pregnancies complicated by poorly controlled diabetes.\n- Option D: Oligohydramnios is not specifically associated with maternal diabetes; polyhydramnios is more common.\n- Option E: Post-term delivery is less likely as maternal diabetes often leads to prematurity.", "Conclude that the fetus is most likely at increased risk for neural tube defects due to the mother's poor glycemic control during early pregnancy.\n\nFinal Answer: C: Neural tube defect."] | ["Identify the patient's condition and history. The patient is a 31-year-old woman with type 1 diabetes mellitus, presenting for her first prenatal visit. Her last menstrual period was 8 weeks ago, indicating early pregnancy. Her hemoglobin A1c is 13.7%, indicating poor glycemic control. She also has microalbuminuria, suggesting longstanding diabetes-related complications.", "Understand the implications of the patient's medical condition. Poor glycemic control in diabetes, particularly during the periconceptional period, increases the risk of congenital malformations in the fetus. High hemoglobin A1c levels and microalbuminuria further suggest inadequate management of diabetes, posing a higher risk for fetal complications.", "Identify the specific risks associated with maternal diabetes. Infants of diabetic mothers (IDM) are at increased risk for several congenital anomalies and complications. Common congenital malformations include cardiovascular defects (e.g., septal defects, transposition of the great arteries) and neural tube defects (e.g., anencephaly, spina bifida). Other risks include sacral agenesis and small left colon syndrome.", "Recognize the most likely complications in this scenario. Given the patient's poorly controlled diabetes, the fetus is at increased risk for neural tube defects. These defects are specifically mentioned in the reasoning process and are a well-known risk associated with maternal hyperglycemia.", "Conclusion. The fetus is most likely at increased risk for neural tube defects due to the mother's poorly controlled type 1 diabetes mellitus and elevated hemoglobin A1c during early pregnancy."] |

Table 8: Composition of MedReason-Dx.

| source | MCQ | QEQ |
|---|---|---|
| JAMA Clinical Challenge | 210 | 204 |
| Medbullets | 197 | 125 |
| MMedBench | 164 | 35 |
| nephSAP | 21 | 7 |
| LiveQA | 0 | 94 |
| PubMedQA | 0 | 113 |
| total | 592 | 578 |

Table 9: A piece of data from MedReason-Dx.

| id | 9cb1a646-3e55-5b96-a769-b06289709164 |
|---|---|
| **specialty** | Pediatrics |
| **question** | A 6-year-old boy presents to the emergency department after falling from his scooter. He has dull, aching pain along his left side where he hit the ground. He fell sideways off the scooter as he rounded a curve in the road. He has never had any serious injuries but that he always seems to bruise easily, especially after he started playing soccer this fall. His parents deny that he has an abnormal number of nosebleeds or bleeding from the gums. They have never seen blood in his stool or urine. His mother notes that her brother has had similar problems. His temperature is 98.6°F (37°C), blood pressure is 112/74 mmHg, pulse is 82/min, and respirations are 11/min. On physical exam, the patient has extensive bruising of the lateral left thigh and tenderness to palpation. Laboratory tests are performed and reveal the following:\n\nHemoglobin: 14 g/dL\nHematocrit: 41%\nMean corpuscular volume: 89 μm3\nReticulocyte count: 0.8%\nLeukocyte count: 4,700/mm3\nProthrombin time (PT): 13 seconds\nPartial thromboplastin time (PTT): 56 seconds\nBleeding time (BT): 4 minutes\n\nWhich of the following is the most likely underlying pathophysiology? |
| **answer_options** | ["Anti-platelet antibodies", "Factor 8 deficiency", "Factor 9 deficiency", "GP1b deficiency", "Von Willebrand factor deficiency"] |
| **answer** | Factor 8 deficiency |
| **answer_idx** | B |
| **answer_steps** | ["A 6-year-old boy presents with extensive bruising after falling, and has a family history of similar bleeding problems, suggesting a possible hereditary bleeding disorder.", "The laboratory findings show a prolonged partial thromboplastin time (PTT) with normal prothrombin time (PT) and bleeding time (BT), which suggests a coagulation factor deficiency rather than a platelet function disorder.", "The isolated prolongation of PTT with normal PT and BT is characteristic of hemophilia, which can be caused by either factor 8 deficiency (hemophilia A) or factor 9 deficiency (hemophilia B).", "Hemophilia A (factor 8 deficiency) and hemophilia B (factor 9 deficiency) present with similar symptoms, including hemarthrosis, frequent bruising, and prolonged PTT. However, hemophilia A is more common than hemophilia B.", "Given the epidemiological prevalence and the clinical presentation, factor 8 deficiency (hemophilia A) is the most likely diagnosis.", "Therefore, the most likely underlying pathophysiology is factor 8 deficiency, making 'B': Factor 8 deficiency the correct answer."] |
| **answer_key points** | ["extensive bruising", "family history", "prolonged partial thromboplastin time", "normal prothrombin time", "normal bleeding time", "hemophilia A", "factor 8 deficiency", "intramuscular hematoma", "gastrointestinal bleeding", "isolated prolongation of PTT", "normal bleeding time", "factor replacement", "anti-factor antibodies"] |

Table 10: Response from DeepSeek R1.

| model | reasoning steps | key points |
|-------|-----------------|------------|
| DeepSeek R1 | ["The patient presents with **prolonged PTT (56 seconds)**, normal PT (13 seconds), and **normal bleeding time (4 minutes)**. This suggests a **factor deficiency in the intrinsic coagulation pathway** (e.g., Factor VIII, IX) rather than platelet dysfunction or extrinsic pathway issues.", "**Bleeding time is normal**, ruling out primary platelet disorders (e.g., Bernard-Soulier syndrome, ITP) and mild von Willebrand disease (vWD), where bleeding time is often prolonged due to impaired platelet adhesion.", "**Family history** (maternal uncle with similar symptoms) suggests an **X-linked recessive disorder**, narrowing the diagnosis to **Hemophilia A (Factor VIII deficiency)** or **Hemophilia B (Factor IX deficiency)**. Both cause prolonged PTT with normal PT and bleeding time.", "**No mucosal bleeding** (denied nosebleeds, gum bleeding) and trauma-related bruising align more with **Hemophilia** than vWD, which commonly involves mucosal bleeding.", "**Hemophilia A (Factor VIII deficiency)** is **more prevalent** than Hemophilia B, and the X-linked inheritance pattern fits the maternal uncle's history.\n\nTherefore, the answer is \\box{B}."] | ["'prolonged PTT (56 seconds)'", "'normal PT (13 seconds)'", "'normal bleeding time (4 minutes)'", "'factor deficiency in the intrinsic coagulation pathway'", "'Bleeding time is normal'", "'ruling out primary platelet disorders'", "'Family history'", "'X-linked recessive disorder'", "'Hemophilia A (Factor VIII deficiency)'", "'Hemophilia B (Factor IX deficiency)'", "'prolonged PTT with normal PT and bleeding time'", "'No mucosal bleeding'", "'trauma-related bruising'", "'Hemophilia'", "'Hemophilia A (Factor VIII deficiency)'", "'more prevalent'", "'X-linked inheritance pattern'"] |

Table 11: Response from GPT-4o.

| model | reasoning steps | key points |
|-------|-----------------|------------|
| GPT-4o | ["Analyze the clinical presentation. The 6-year-old boy presents with easy bruising but no significant mucosal bleeding (such as nosebleeds or gum bleeding), which suggests a clotting factor deficiency rather than a platelet function disorder.", "Review the family history. The maternal uncle has had similar bleeding issues, suggesting a potential X-linked recessive pattern of inheritance, which is typical for hemophilia.", "Examine the laboratory findings. The prolonged partial thromboplastin time (PTT) with normal prothrombin time (PT) and bleeding time (BT) suggest an intrinsic pathway clotting factor deficiency.", "Consider the specific clotting factor deficiencies that could cause the prolonged PTT. Hemophilia A (Factor 8 deficiency) and Hemophilia B (Factor 9 deficiency) are the two most common intrinsic pathway deficiencies. Both would cause a prolonged PTT without affecting PT or BT.", "Choose between Factor 8 and Factor 9 deficiency. While both conditions fit the laboratory pattern, Hemophilia A (Factor 8 deficiency) is more common than Hemophilia B (Factor 9 deficiency).\n\nTherefore, the answer is \\box{B}."] | ["6-year-old boy", "easy bruising", "no significant mucosal bleeding", "clotting factor deficiency", "family history", "maternal uncle", "X-linked recessive pattern", "hemophilia", "prolonged partial thromboplastin time (PTT)", "normal prothrombin time (PT)", "normal bleeding time (BT)", "intrinsic pathway", "Hemophilia A", "Factor 8 deficiency", "Hemophilia B", "Factor 9 deficiency", "prolonged PTT", "more common", "answer is B"] |

Table 12: Response from QWQ 32B.

| model | reasoning steps | key points |
|-------|-----------------|------------|
| QWQ 32B | ["Step 1: **Identify key clinical findings**: The patient presents with easy bruising, a family history of similar issues (mother's brother), and normal hemoglobinhematocrit", "The labs show a normal PT, prolonged PTT (56 seconds), and a normal bleeding time (4 minutes)", "Step 2: **Analyze coagulation studies**: \n- **Normal PT** indicates the extrinsic pathway (factors VII, X, etc", ") is intact", "- **Prolonged PTT** suggests a defect in the intrinsic or common pathway", "- **Normal bleeding time** rules out platelet dysfunction (e", "g", ", von Willebrand disease, Bernard-Soulier syndrome, or anti-platelet antibodies)", "Step 3: **Consider differentials for prolonged PTT with normal PT and bleeding time**:\n- **Factor deficiencies in the intrinsic pathway**: Factors VIII (hemophilia A), IX (hemophilia B), or XI", "- **Factor VIII deficiency (hemophilia A)** and **Factor IX deficiency (hemophilia B)** are X-linked recessive disorders, fitting the family history (mother's brother affected)", "- **Factor XI deficiency** is autosomal recessive and less likely given the family history", "Step 4: **Differentiate between hemophilia A and B**:\n- Both cause prolonged PTT and bruising/bleeding but are clinically similar", "- **Hemophilia A (Factor VIII deficiency)** is more common than hemophilia B", "- The family history (maternal uncle affected) suggests an X-linked disorder, compatible with either", "However, hemophilia A is more prevalent", "Step 5: **Exclude other options**:\n- **Von Willebrand disease (E)**: Causes prolonged PTT and bleeding time", "However, the normal bleeding time here makes this less likely", "- **GP1b deficiency (D)**: Causes prolonged bleeding time and platelet dysfunction, which is not seen here", "- **Anti-platelet antibodies (A)**: Would cause thrombocytopenia or abnormal platelet function, but bleeding time is normal", "- **Factor IX deficiency (C)** is possible but less common than hemophilia A", "Step 6: **Conclusion**: The prolonged PTT, normal PT and bleeding time, family history of X-linked bleeding, and higher prevalence of hemophilia A make **Factor 8 deficiency (hemophilia A)** the most likely diagnosis", "Therefore, the answer is \\boxed{B}"] | ["key clinical findings", "easy bruising", "family history", "normal hemoglobin", "hematocrit", "normal PT", "prolonged PTT", "normal bleeding time", "platelet dysfunction", "factor deficiencies", "intrinsic pathway", "hemophilia A", "factor VIII deficiency", "hemophilia B", "family history", "X-linked recessive disorders", "maternal uncle", "exclude other options", "von Willebrand disease", "GP1b deficiency", "anti-platelet antibodies", "factor IX deficiency", "conclusion"] |

Table 13: Response from PMC LLaMA 13B.

| model | reasoning steps | key points |
|-------|-----------------|------------|
| PMC LLaMA 13B | ["With these laboratory values, it is clear there is a problem with the bone marrow production of platelets, as demonstrated by an extremely low reticulocyte count and an increased mean corpuscular volume, which is indicative of a megaloblastic process", "This child likely has a disorder involving the thrombopoietin receptor, which is essential for megakaryocyte growth and maturation and for the release of platelets from the marrow", "The gene for the thrombopoietin receptor is located on chromosome 11", "Most cases of congenital amegakaryocytic thrombocytopenia result from autosomal recessive inheritance of a nonsense point mutation involving one of the thrombopoietin receptor genes", "The gene for the thrombopoietin receptor is the same as that associated with congenital amegakaryocytic thrombocytopenia", "This autosomal dominant form is extremely rare", "The other choices are not associated with a low reticulocyte count and a giant platelet", "Patients with antiplatelet antibodies, or with factor 8, 9, or Von Willebrand factor deficiencies have normal to increased platelet counts", "Patients with GP1b deficiency usually have a normal or increased platelet count, but a dysfunctional platelet population", "###Answer: OPTION D IS CORRECT"] | ["laboratory values", "bone marrow production", "platelets", "low reticulocyte count", "increased mean corpuscular volume", "megaloblastic process", "thrombopoietin receptor", "megakaryocyte growth", "maturation", "release of platelets", "chromosome 11", "congenital amegakaryocytic thrombocytopenia", "autosomal recessive inheritance", "nonsense point mutation", "autosomal dominant form", "normal to increased platelet counts", "dysfunctional platelet population"] |

