# OpenReview forum: "MedReason-Dx: Benchmarking Step-by-Step Reasoning of Language Models in Medical Diagnosis"
_ICLR.cc/2026/Conference — Submitted to ICLR 2026_

### Official Review · Reviewer_NaHL · 2025-10-19

**Soundness:** 2
**Presentation:** 3
**Contribution:** 2
**Rating:** 6
**Confidence:** 3

**Summary:**

This paper introduces MedReason-Dx, a good benchmark designed to address the critical gap in evaluating step-by-step reasoning processes of language models (LLMs) in medical diagnosis—an aspect largely overlooked by existing medical QA benchmarks that focus solely on final answer accuracy. Experiments on LLMs reveal three core findings: 1) State-of-the-art LLMs struggle with complex medical reasoning; 2) LLMs face challenges in recalling all essential medical knowledge; 3) Medical LLMs do not outperform general-purpose LLMs in complex reasoning tasks, attributed to insufficient training on reasoning-focused datasets.

**Strengths:**

1. MedReason-Dx fills a critical niche by focusing on step-by-step reasoning in medical diagnosis, a dimension ignored by dominant medical QA benchmarks.
2. The five evaluation metrics provide a holistic assessment of reasoning, and experiments on 11 LLMs (covering different sizes and domains) ensure generalizable findings.
3. The paper explains technical concepts in accessible terms, with formulas and examples to illustrate key ideas.
4. The work advances the development of trustworthy medical AI by emphasizing interpretability—essential for clinical adoption.

**Weaknesses:**

1. Table 1 is missing data for critical columns (e.g., CoT Evaluation, Expert Annotation) across existing benchmarks, making it difficult to fully assess MedReason-Dx’s uniqueness relative to prior work like MedReason (Wu et al., 2025a) and MedCaseReasoning (Wu et al., 2025b).
2. The paper does not report how consistent expert annotators were in creating reasoning chains and key points. This raises questions about the reliability of the ground truth, as subjective differences in clinical reasoning could introduce bias.
3. The experiments identify performance gaps but do not delve into why medical LLMs underperform or how to mitigate these issues (e.g., fine-tuning strategies using the benchmark).
4. Converting multiple-choice questions to open-ended formats using LLMs, even with expert review, may introduce subtle changes to clinical context or reasoning requirements. So how should the benchmark control the quality of the dataset?
5. The paper focuses on benchmarking existing models but does not demonstrate how MedReason-Dx can be used to improve LLMs. A good explanation is well recommended.

**Questions:**

In addition to the above concerns, I also have the following questions:
1. Did you compute inter-annotator agreement for the expert-curated reasoning chains and key points? If so, what were the results, and how did you resolve discrepancies? If not, how do you ensure the reliability of the ground truth?
2. Why do direct prompts outperform CoT prompts for open-ended questions in some top-performing models (e.g., GPT-4o)? Is this due to CoT introducing redundant steps or the open-ended format requiring more concise reasoning?
3. How does MedReason-Dx handle rare diseases or niche clinical scenarios?
4. Have you tested MedReason-Dx on LLMs specifically fine-tuned for medical reasoning tasks?

---

> ### Author Response · Authors · 2025-11-24
>
> >Q1: Table 1 is missing data for critical columns across existing benchmarks, making it difficult to fully assess MedReason-Dx’s uniqueness relative to prior work like MedReason and MedCaseReasoning.
>
> A1: We thank the reviewer for the valuable suggestion. To more clearly position MedReason-Dx against closely related benchmarks, we have expanded Table 1 and the discussion in Section 4.3 with the following clarified distinctions:
>
> - MedReason relies on structured medical knowledge graphs to synthesize large-scale reasoning chains. While this approach enables excellent scalability, the resulting tasks are predominantly knowledge-retrieval oriented and may introduce synthesis artifacts, limiting their ability to capture the ambiguity and multi-step clinical reasoning inherent in real-world practice.
>
> - MedCaseReasoning  extracts authentic physician-written reasoning statements from openly accessible case reports, thereby offering significantly higher clinical fidelity than synthetic datasets. However, its evaluation remains restricted to final diagnostic accuracy and coarse-grained reasoning recall.
>
> In contrast, MedReason-Dx is constructed exclusively from real clinical vignettes and challenge cases that explicitly require multi-step diagnostic reasoning, spanning 24 major medical specialties in both multiple-choice and open-ended formats. Each instance is augmented with manually curated ground-truth reasoning chains and **fine-grained** domain-specific medical key points by board-certified specialists. This design supports a substantially more granular evaluation framework comprising five complementary metrics:
>
> - Accuracy (final diagnostic correctness)
>
> - Reasoning Completeness Score (RCS) and Reasoning Necessity Score (RNS): whether all necessary steps are present and superfluous steps are absent
>
> - Key-point Completeness Score (KCS) and Key-point Necessity Score (KNS): precise alignment with clinically essential knowledge elements
>
> These metrics collectively provide a rigorous quantification of reasoning quality, fidelity, and conciseness—dimensions that are not assessed in MedReason or MedCaseReasoning. Consequently, MedReason-Dx offers a more comprehensive and clinically meaningful assessment of LLMs’ true diagnostic reasoning capabilities beyond mere recall or accuracy.
>
> We believe these additions and the revised Table 1 directly address the reviewer’s concern and strengthen the justification for MedReason-Dx’s uniqueness.
>
> >Q2: The paper does not report how consistent expert annotators were in creating reasoning chains and key points. This raises questions about the reliability of the ground truth, as subjective differences in clinical reasoning could introduce bias.
>
> A2: Thanks for the reviewer's comment. Regarding the inter-annotator reliability for the reasoning chains, we deliberately adopted a "Draft-Verification-Adjudication" protocol (Iterative Curation) rather than independent parallel annotation. This methodological decision is grounded in the intrinsic nature of our task and aligns with established academic and clinical benchmarks:
>
> 1. Infeasibility of Exact Agreement for Open-Ended Reasoning Unlike closed-ended classification tasks or mathematical problems where a single canonical answer exists, generating clinical reasoning chains is an open-ended natural language generation task. Due to high linguistic variability, two experts may formulate valid but distinct reasoning paths for the same diagnosis. Consequently, standard inter-annotator agreement metrics (such as Cohen's Kappa or textual overlap scores) often yield artificially low values for valid variations, failing to reflect the true semantic quality and logical soundness of the data.
>
> 2. To ensure quality without penalizing valid linguistic diversity, we followed the "Writer-Verifier" paradigm employed in complex reasoning benchmarks like StrategyQA [1]. Instead of measuring the textual similarity between independent drafts, we focused on functional verification: a second expert reviews the initial draft to verify that the logic is clinically sound and complete. This approach prioritizes logical correctness (validity) over simple phrasing consistency.
>
> 3. Alignment with Clinical Documentation Standards (MIMIC-CXR)Our progressive workflow (Clinician A drafts $\to$ Clinician B reviews $\to$ Senior Clinician C adjudicates discrepancies) mirrors the standard clinical reporting workflow used to generate the ground-truth free-text radiology reports in datasets like MIMIC-CXR [2]. While categorical labels in some benchmarks may rely on majority voting, the generation of the source medical text itself—which serves as the primary ground truth—traditionally follows this hierarchical supervision model (Resident drafting followed by Attending verification). We believe this consensus-based method provides a more robust guarantee of data fidelity for complex medical documentation than averaging the inputs of independent annotators.

---

> ### Author Response · Authors · 2025-11-24
>
> >Q3: The experiments identify performance gaps but do not delve into why medical LLMs underperform or how to mitigate these issues (e.g., fine-tuning strategies using the benchmark).
>
> A3: We appreciate the reviewers' suggestions. We have conducted a more detailed analysis of the failures in medical LLMs. We found that the majority of failures in medical LLMs stems from **knowledge misapplication and reasoning failures**. Unlike general models, medical LLMs possess medical knowledge but frequently **misapply it in wrong clinical contexts**, generating irrelevant medical concepts while missing critical diagnostic pathways. This suggests that **domain-specific fine-tuning on high-quality medical reasoning benchmarks is essential** to teach models how to correctly apply medical knowledge and follow appropriate clinical reasoning principles. We provide an analysis example as follows:
>
> A 49-year-old immunocompromised patient presented with acute unilateral painless vision loss, retinal vasculitis, and frosted branch angiitis. The correct answer is diagnostic anterior chamber paracentesis and intravitreal foscarnet, but the model predicted "escalate systemic immunosuppression."
>
> **Evaluation Results**: Key point recall: 0.02 (1/49 matched, missing 48 ground truth concepts); Precision: 0.05 (1/19 model-generated key points matched, generating 18 irrelevant medical concepts such as azathioprine, prednisone, and drug interactions).
>
> This case demonstrates **knowledge misapplication**: while the model activated medical knowledge (immunosuppressive agents, drug interactions, contraindications), it applied this knowledge to the wrong clinical context. The model failed to recognize that infectious etiologies must be excluded before modifying immunosuppression—a fundamental clinical reasoning principle. The combination of low recall (missing critical concepts) and zero precision (generating irrelevant concepts) indicates systematic reasoning failures where models follow wrong diagnostic pathways despite having relevant medical knowledge.
>
> These analyses highlight a critical limitation of current medical LLMs: they often misapply medical knowledge to inappropriate clinical contexts and fail to follow structured diagnostic reasoning pathways, even when the correct knowledge is present. This underscores the necessity of fine-tuning on high-quality, expert-verified clinical reasoning datasets to teach models proper knowledge application and clinically valid reasoning patterns.
>
> In the revision, we have added more representative failure to further illustrate this point.
>
>
> >Q4: Converting multiple-choice questions to open-ended formats using LLMs, even with expert review, may introduce subtle changes to clinical context or reasoning requirements. So how should the benchmark control the quality of the dataset?
>
> A4: We thank the reviewer for the comment. In fact, when converting multiple-choice questions into open-ended format, we only paraphrase the final interrogative sentence (typically the last 3–12 words, e.g., “What is the most likely diagnosis?”), while keeping the entire clinical vignette, patient history, examination findings and lab results 100% identical to the original source. We have further emphasized this point in Section 3.1.2.

---

> ### Author Response · Authors · 2025-11-24
>
> >Q5: The paper focuses on benchmarking existing models but does not demonstrate how MedReason-Dx can be used to improve LLMs. A good explanation is well recommended.
>
> A5: We thank the reviewer for this important comment.
> We agree that the current paper focuses on benchmarking and dataset contribution, but does not yet demonstrate tangible LLM improvement via training on MedReason-Dx.
>
> In medicine, creating large-scale, expert-verified reasoning data is extremely difficult and costly. Existing high-quality medical datasets (including ours) remain 1–2 orders of magnitude smaller than general-domain instruction datasets. It is widely observed in the community that conventional supervised fine-tuning or continued pre-training on such limited specialized data typically fails to yield stable and substantial gains on complex diagnostic reasoning tasks.
>
> We believe that realizing genuine performance improvement on such small datasets will require new training paradigms (e.g., iterative self-refinement with expert validation) that can effectively leverage small but high-quality seed data like MedReason-Dx. Designing such methods is highly non-trivial and constitutes our major ongoing and future work.
>
>
> >Q6: Why do direct prompts outperform CoT prompts for open-ended questions in some top-performing models (e.g., GPT-4o)? Is this due to CoT introducing redundant steps or the open-ended format requiring more concise reasoning?
>
> A6: Thank you for the reviewer's insightful comment. Given that chain-of-thought reasoning significantly enhances model performance, advanced models like GPT-4o typically incorporate this capability during training. Explicitly prompting them to “think step by step” appears unnecessary and may even degrade performance by deviating from training-time prompts, leading to overemphasis on formatting.
>
> >Q7: How does MedReason-Dx handle rare diseases or niche clinical scenarios?
>
> A7: MedReason-DX does include a small number of rare diseases and niche clinical scenarios. Examples include Good syndrome, head and neck teratomas, esophageal atresia, factitious disorder and others. While rare diseases are inherently underrepresented by definition, their inclusion reflects real-world clinical practice where physicians must consider both common and uncommon conditions. These cases have also been meticulously annotated by our experts.
>
>
> References:
>
> [1] Did Aristotle Use a Laptop? A Question Answering Benchmark with Implicit Reasoning Strategies.
>
> [2] MIMIC-CXR, a de-identified publicly available database of chest radiographs with free-text reports.

---

> > ### Comment · Reviewer_NaHL · 2025-11-24
> >
> > I believe the authors have solved most of my problems;
> > I will continue to pay attention and refer to the comments of other reviewers to maintain my positive score.

---

### Official Review · Reviewer_EX4N · 2025-10-30

**Soundness:** 2
**Presentation:** 2
**Contribution:** 2
**Rating:** 4
**Confidence:** 4

**Summary:**

The authors introduce MedReason-Dx, a benchmark to evaluate not just answer accuracy but the reasoning process of LLMs on medical diagnosis questions. The dataset spans 1,170 items across 24 specialties, each with expert-annotated, stepwise solutions and extracted key points. They propose five metrics to quantify reasoning fidelity. Baselines across general and medical LLMs show modest accuracies, with general models typically outperforming medical LLMs on complex reasoning. Evaluation relies on LLM-as-judge (GPT-4o-mini), with a small cross-judge consistency check vs. GPT-4.1-mini.

**Strengths:**

- This work is well motivated to evaluate the quality of reasoning paths in medical diagnosis.
- The proposed benchmark covers a broad range of specialities, with both multiple-choice and open-ended questions for comprehensive evaluation.
- Along with the benchmark questions, this work also proposes various reasoning-focused metrics (RCS/RNS/KCS/KNS) to evaluate reasoning quality from different perspectives.
- Comprehensive evaluations are conducted on a broad range of LLMs, providing insights into their differences in medical reasoning.

**Weaknesses:**

- My major concern is about the potential existence of multiple reasoning paths for the same medical question. For example, if concept A connects to B and B to C, but there is also a direct link from A to C, it is unclear how MedReason-Dx accounts for such alternative but correct reasoning routes.
- It is surprising that DeepSeek R1 has a higher accuracy than V3 (both multiple-choice and open-ended) but shows a lower reasoning quality (e.g., lower RNS/RCS on multiple-choice questions, RNS/RCS/KCS on open-ended questions). A deeper analysis beyond the scores is needed to explain why such an inconsistency can happen and if the evaluation metrics for reasoning are reliable.
- While ablation of LLM judges is conducted on GPT-4o-mini and GPT-4.1-mini by comparing their evaluation scores, there is no analysis on how well they perform on the given task. Given the relatively low accuracy of all LLMs on the constructed benchmark (Table 5), it is unclear if GPT-4o-mini/GPT-4.1-mini is capable of doing such evaluations.
- The average response lengths (or number of reasoning steps / key points) also need to be reported for the evaluated LLMs, which may have a significant impact on the reasoning-related metrics

**Questions:**

See the listed weaknesses.

---

> ### Author Response · Authors · 2025-11-24
>
> >Q1: My major concern is about the potential existence of multiple reasoning paths for the same medical question. For example, if concept A connects to B and B to C, but there is also a direct link from A to C, it is unclear how MedReason-Dx accounts for such alternative but correct reasoning routes.
>
> A1: We sincerely thank the reviewer for this thoughtful and important comment regarding the potential for multiple reasoning paths.
>
> We agree that in broad reasoning tasks, divergence is common. However, we argue that **clinical diagnostic reasoning is uniquely constrained by the "standard of care,"** which enforces a high degree of convergence on the core reasoning backbone.
>
> **1. Convergence in Clinical Safety and Guidelines.**
> Unlike creative writing or open-ended logic, clinical reasoning requires adherence to evidence-based protocols. While minor variations in wording or the order of gathering non-critical information naturally occur, the **substantive logical backbone** must remain consistent to ensure patient safety.
>
> A classic illustration is the evaluation of a patient with a sudden, severe “thunderclap” headache. While one clinician might loosely entertain migraine or sinusitis, and another might focus on aneurysm rupture immediately, **only the reasoning path that rapidly recognizes the red-flag pattern and prioritizes ruling out Subarachnoid Hemorrhage (SAH)** is considered clinically acceptable. A reasoning chain that spends multiple steps exploring benign causes before addressing the life-threatening possibility is not an "equally valid alternative"; in medical boards and malpractice reviews, it is explicitly criticized as dangerously delayed and below the standard of care. Our reference chains, constructed by senior attending physicians, capture precisely this **non-negotiable safety-critical core**.
>
> **2. Tolerance via Key Point Evaluation (KCS/KNS).**
> To address the reviewer's specific concern about valid variations (e.g., a direct link $A \to C$ vs. mediated $A \to B \to C$), our evaluation metric is designed to be permissive at the stylistic level. The Key Point (KCS/KNS) layer focuses on the presence of **indispensable semantic elements** rather than rigid syntactic matching.
> * If step $B$ is a trivial connector or stylistic filler, our Key Points likely only require $A$ and $C$, effectively crediting both the direct and mediated paths.
> * If step $B$ is a **crucial intermediate pathophysiological step** (e.g., a specific lab finding required to justify the diagnosis), then skipping it ($A \to C$) would be considered an "unexplained jump" in a pedagogical context, and is rightly penalized for lacking interpretability.
>
> **3. Alignment with Broader Benchmarking Standards.**
> Finally, we note that single-reference evaluation remains the standard practice in virtually all large-scale reasoning benchmarks (e.g., GSM8K, MATH, AIME), even in domains like mathematics where alternative proofs exist. Given the stronger convergence requirements of medicine compared to general logic, we believe this approach is statistically robust for benchmarking purposes.
>
> In the revised manuscript, we have added a discussion in the **Limitations section ** to explicitly acknowledge the theoretical possibility of greater divergence in rare, ambiguous cases, while clarifying why the single-reference "standard of care" approach is appropriate for the vast majority of diagnostic scenarios.

---

> ### Author Response · Authors · 2025-11-24
>
> >Q2: It is surprising that DeepSeek R1 has a higher accuracy than V3 but shows a lower reasoning quality. A deeper analysis beyond the scores is needed to explain why such an inconsistency can happen and if the evaluation metrics for reasoning are reliable.
>
> A2: We thank the reviewer for this insightful and important observation. The higher diagnostic accuracy of DeepSeek R1 compared to DeepSeek-V3, despite its lower reasoning quality scores, indeed appears counter-intuitive at first glance. Below we provide a more in-depth analysis grounded in recent findings in the literature.
>
> Recent work has systematically demonstrated that current reinforcement learning from verifiable rewards (RLVR)—the primary post-training technique used in DeepSeek R1 and similar reasoning models—**primarily improves sampling efficiency (i.e., pass@1) rather than expanding the model’s fundamental reasoning capacity** beyond that of the underlying base model [1]. In other words, RLVR increases the probability of sampling high-quality reasoning paths that already exist in the base model’s output distribution, but rarely discovers qualitatively novel reasoning strategies.
>
> This mechanism directly explains the observed discrepancy in MedReason-Dx:
>
> 1. **Accuracy gains originate from improved sampling efficiency**: DeepSeek R1 (built on DeepSeek-V3 + large-scale RLVR on mathematics and coding) is heavily optimized to surface correct solutions in a single generation. On MEDREASON-DX, this manifests as higher final diagnostic accuracy.
>
> 2. **Reasoning quality metrics (RNS/RCS/KCS) are designed to measure alignment with expert-defined clinical reasoning structures**, not merely outcome correctness. DeepSeek R1, lacking targeted post-training on clinical reasoning datasets, tends to produce longer, exploratory, and sometimes circuitous reasoning traces that deviate from the concise, hypothesis-driven patterns taught in medical education. Such traces frequently receive lower step-wise scores from both automated metrics and clinician evaluators, even when the final diagnosis is correct.
>
> 3. **Domain transfer limitation of current reasoning models**: As shown in Yue et al. [1], reasoning capabilities induced by RLVR on mathematical/coding tasks exhibit limited generalization to structurally different domains such as clinical reasoning, where background medical knowledge and specific diagnostic frameworks (e.g., illness scripts, Bayesian reasoning) play a critical role. Consequently, R1’s “generic” reasoning strategies—while effective for convergence—are poorly calibrated to the evaluation criteria embedded in RNS/RCS/KCS.
>
> Far from indicating unreliability of our proposed metrics, this discrepancy constitutes a **key finding** of our study: **answer correctness and reasoning process quality can decouple significantly** in contemporary reasoning models when applied out-of-distribution. Models may achieve high accuracy via compensatory mechanisms (extended search in the latent space of the base model) while producing reasoning traces that clinicians judge as suboptimal or educationally inappropriate.
>
> In summary, the seemingly paradoxical results for DeepSeek R1 are consistent with the current theoretical understanding of RLVR: it enhances efficiency within the base model’s capability envelope but does not reliably improve— and may even degrade—structured reasoning quality in specialized domains without domain-specific alignment. We believe this observation significantly strengthens the paper’s core message about the limitations of current general-purpose reasoning models in high-stakes clinical settings.

---

> ### Author Response · Authors · 2025-11-24
>
> >Q3: While ablation of LLM judges is conducted on GPT-4o-mini and GPT-4.1-mini by comparing their evaluation scores, there is no analysis on how well they perform on the given task. Given the relatively low accuracy of all LLMs on the constructed benchmark (Table 5), it is unclear if GPT-4o-mini/GPT-4.1-mini is capable of doing such evaluations.
>
> A3: We acknowledge this concern. However, **the task required of LLM judges is fundamentally different from and significantly simpler than the medical reasoning task itself**.
>
> LLM judges assess **semantic similarity between pairs of medical reasoning steps or medical terminology**, rather than performing complex multi-step diagnostic reasoning. This similarity assessment task is substantially simpler than the full diagnostic reasoning evaluated in our benchmark—while medical reasoning requires synthesizing complex clinical information and multi-step inference, similarity assessment primarily requires semantic understanding and pattern matching, which is well within the capabilities of GPT-4o-mini.
>
> To address this concern, we conducted additional experiments using multiple LLM judges (GPT-4o-mini, GPT-4.1-mini, and DeepSeek). Our results show that **different LLM judges yield highly consistent evaluation scores**, demonstrating both that the evaluation task is within the models' capabilities and that our evaluation process is stable and reliable. The observed low accuracy in Table 5 reflects genuine limitations in medical reasoning capabilities, not limitations in our evaluation methodology. To further validate the reliability of our evaluation protocol, we randomly selected 50 questions and invited two senior clinicians to independently assess the model-generated reasoning processes. Following the exact same criteria as our automated judges, they marked each reasoning step, from which F1 scores were computed. The Pearson correlations of these F1 scores consistently exceed 0.8 across all pairs of human and LLM judges. These results demonstrate strong agreement between automated and human assessment, confirming the effectiveness of our evaluation process.
>
> In the revised version, we have added these additional experimental results to the corresponding tables.
>
> | Judge         | GPT-4o          |          | DeepSeek R1     |          |
> |---------------|-----------------|------------------|-----------------|------------------|
> |               | RNS    | RCS    | RNS    | RCS    |
> | GPT-4o-mini   | 89.92  | 69.76  | 86.11  | 73.46  |
> | GPT-4.1-mini  | 88.47  | 72.63  | 88.08  | 73.83  |
> | Claude-Haiku-4.5  | 83.38  |  65.85 | 83.78  | 71.16  |
> | DeepSeek-V3.2-Exp  | 80.23  |  67.74 | 81.02  | 74.39  |
>
> >Q4: The average response lengths (or number of reasoning steps / key points) also need to be reported for the evaluated LLMs, which may have a significant impact on the reasoning-related metrics
>
> A4: Following the reviewer’s valuable suggestion, we have reported the average number of extracted key points and reasoning steps for all evaluated models in the revision. The results have been added to the revision.
>
> | Model | Multiple-choice (Key Points) | Multiple-choice (Reasoning Steps) | Open-ended (Key Points) | Open-ended (Reasoning Steps) |
> |------|------------------|----------------|------------------|----------------|
> | BioMistral-7B | 18.06  | 6.08 | 15.95  | 6.52  |
> | HuatuoGPT2-7B | 21.50 | 3.67 | 24.10 | 4.41  |
> | PMC_LLaMA_13B | 22.34 | 19.40 | 20.29 | 8.66 |
> | Baichuan4_Turbo | 21.33  | 5.11 | 25.28 | 5.50  |
> | GPT-4o | 21.34 | 4.84 | 25.01 | 5.60 |
> | Llama-3.3-70B | 26.04 | 16.48 | 27.90 | 15.64 |
> | Mistral_7B | 20.93 | 8.80 | 22.35 | 9.39 |
> | o1_mini | 21.55 | 4.75 | 26.21 | 5.65 |
> | QwQ_32B | 27.91 | 11.62 | 28.67 | 10.61 |
> | DeepSeek-R1 | 25.13 | 4.77 | 27.97 | 5.01 |
> | DeepSeek-V3 | 25.33 | 4.68 | 29.87 | 5.15 |
>
>
> Reference:
>
> [1]: Does Reinforcement Learning Really Incentivize Reasoning Capacity in LLMs Beyond the Base Model?

---

### Official Review · Reviewer_3aPJ · 2025-10-30

**Soundness:** 3
**Presentation:** 4
**Contribution:** 3
**Rating:** 6
**Confidence:** 3

**Summary:**

This paper designed a benchmarking system to evaluate medical step-by-step reasoning in LLMs. Instead of only aiming for the final correct option, the authors, along with an accuracy metric, propose evaluation metrics based on reasoning chains and key point based evaluation to assess the completeness and necessity of the generated reasoning steps. The authors construct a human-annotated dataset with 1170 questions derived from multiple choice and open-ended question datasets spanning across 24 topics.

**Strengths:**

- **Expert annotated dataset** to evaluate step-by-step reasoning in the medical domain which would benefit in understanding and evaluating the decision-making of models.
- **Experimental setup**. Including both general-purpose and domain-specific LLMs helps to clearly see the gaps in performance.
- **Clarity**. The paper is well-structured and easy to follow

**Weaknesses:**

- Although the process of data curation is well explained, it remains unclear which datasets exactly where used to construct MedReason-Dx, the paper could benefit from pointing out which datasets the authors relied on.
- **Lack of information on human experts**. No clear information on the number of human annotators and the annotation agreement at every step where humans were involved, hence, difficult to derive the results for any bias.
- The paper heavily emphasizes quantitative metrics but provides limited qualitative discussion of failure cases or examples where LLMs’ reasoning diverges from human reasoning. Such insights would clarify *why* models fail, especially for the medical reasoning tasks.
- Using LLM-judges only from one provider (GPT-4o-mini, GPT-4.1-mini) does not allow to conclude that the evaluation is robust and fair. Perhaps authors should have considered judges from other families, including open models.

**Questions:**

* How many human annotators were involved in the annotation process? How were they selected and what was the inter-annotator agreement?
* Address the issues listed in the weaknesses

---

> ### Author Response · Authors · 2025-11-24
>
> >Q1: Although the process of data curation is well explained, it remains unclear which datasets exactly where used to construct MedReason-Dx, the paper could benefit from pointing out which datasets the authors relied on.
>
> A1: We appreciate the reviewer's suggestions and have further clarified the data sources as follows:
>
> | source                  | # Multiple-Choice Questions   | # Open-Ended Questions   |
> |-------------------------|-------|-------|
> | JAMA Clinical Challenge | 210   | 204   |
> | Medbullets              | 197   | 125   |
> | MMedBench               | 164   | 35    |
> | nephSAP                 | 21    | 7     |
> | LiveQA                  | 0     | 94    |
> | PubMedQA                | 0     | 113   |
> | **total**               | **592** | **578** |
>
> We have further clarified this information in Table 8.
>
> >Q2: Lack of information on human experts. No clear information on the number of human annotators and the annotation agreement at every step where humans were involved, hence, difficult to derive the results for any bias.
>
> A2: We sincerely thank the reviewer for highlighting the importance of transparency regarding our annotation team and quality control process. We have explicitly detailed the expert qualifications and the rationale behind our bias mitigation strategy in the revision:
>
> 1. Expert Qualifications and Hierarchical Workflow. To ensure clinical accuracy, we strictly employed board-certified clinicians rather than crowd-sourced workers. For every single data entry, the curation process involved a hierarchical workflow of up to three independent experts:
>
> Drafting: An initial clinician drafts the reasoning chain.
>
> Verification: A second, independent expert reviews the logical soundness.
>
> Adjudication: In cases of disagreement, a third senior expert adjudicates and finalizes the content. This ensures that every sample represents a multi-expert consensus rather than the subjective opinion of a single individual.
>
> 2. Rationale for Method Selection (Writer-Verifier Paradigm) We deliberately adopted a "Draft-Verification-Adjudication" protocol rather than calculating standard inter-annotator agreement (IAA) metrics. This methodological decision is grounded in the open-ended nature of our task. As noted in the "Writer-Verifier" paradigm [1], high linguistic variability in complex reasoning means that two experts often formulate valid but textually distinct explanations. Standard metrics (e.g., Cohen’s kappa) penalize these valid stylistic variations. Therefore, we focused on functional verification (ensuring the logic is clinically sound) rather than textual overlap.
>
> 3. Bias Mitigation via Hierarchical Supervision Regarding the concern on bias, we address this through Hierarchical Supervision rather than simple voting. Unlike independent annotation schemes that may merely average the biases of multiple individuals, our multi-stage review process explicitly identifies and corrects subjective errors or idiosyncratic reasoning at each step. This workflow aligns with the standard clinical reporting mechanism used to generate the ground-truth free-text reports in datasets like MIMIC-CXR [2], ensuring that the final dataset represents a verified clinical standard.

---

> ### Author Response · Authors · 2025-11-24
>
> >Q3: The paper heavily emphasizes quantitative metrics but provides limited qualitative discussion of failure cases or examples where LLMs’ reasoning diverges from human reasoning. Such insights would clarify why models fail, especially for the medical reasoning tasks.
>
> A3: We acknowledge the importance of qualitative analysis in understanding model failures. Through detailed examination of failure cases, we identify a critical limitation: **LLMs tend to make decisions based on partial feature matching, while human experts engage in comprehensive knowledge integration**. This fundamental difference in reasoning approaches explains many model failures in medical reasoning tasks.
>
> To illustrate this divergence, we present a representative case: A 1-week-old newborn presented with labored breathing, xerostomia, and poor bowel movements. MRI revealed a midline sublingual multiloculated cystic lesion that was T2-hyperintense, T1-hypointense, with no enhancement, fat, or calcifications. The **ground truth diagnosis** was cystic teratoma (mature), confirmed by pathology. The **model (GPT-4o) incorrectly predicted** macrocystic lymphatic malformation.
>
> The model's reasoning focused on surface-level imaging feature matching, concluding that "*Cystic teratoma (mature) usually contains fat, calcifications...which are not present here*" — **treating the absence of these features as exclusionary evidence**. In contrast, the human expert explicitly stated: "*However, absence of calcifications or fat does not rule out teratoma, especially given the midline location and multiloculated cystic nature.*" The expert integrated multiple knowledge domains: (1) fundamental concepts (teratomas as congenital tumors from all three germ layers, incidence 1:20,000-1:40,000), (2) clinical knowledge (head and neck teratomas account for 3-5% of pediatric cases, characteristic midline location), (3) diagnostic principles (pathology as gold standard), and (4) clinical significance (airway obstruction risk, 80-100% mortality if untreated).
>
> This case reveals that **model failures stem from fundamentally different reasoning paradigms**: pattern matching versus knowledge integration. Models excel at identifying isolated features but struggle to apply diagnostic principles (e.g., "absence of evidence is not evidence of absence") and to integrate multi-domain knowledge contextually — capabilities that define expert clinical reasoning.
>
> We have provided more similar analyses in the Appendix.
>
>
> >Q4: Using LLM-judges only from one provider (GPT-4o-mini, GPT-4.1-mini) does not allow to conclude that the evaluation is robust and fair. Perhaps authors should have considered judges from other families, including open models.
>
> A4: We appreciate the reviewer's suggestions. To address the concern about judge diversity and robustness, we evaluated using four different judges from three model families (OpenAI, Anthropic, DeepSeek). As shown below, score remains highly stable, and variance is small, confirming the fairness of our evaluation protocol.
>
> | Judge         | GPT-4o          |          | DeepSeek R1     |          |
> |---------------|-----------------|------------------|-----------------|------------------|
> |               | RNS    | RCS    | RNS    | RCS    |
> | GPT-4o-mini   | 89.92  | 69.76  | 86.11  | 73.46  |
> | GPT-4.1-mini  | 88.47  | 72.63  | 88.08  | 73.83  |
> | Claude-Haiku-4.5  | 83.38  |  65.85 | 83.78  | 71.16  |
> | DeepSeek-V3.2-Exp  | 80.23  |  67.74 | 81.02  | 74.39  |
>
>
> References:
>
> [1] Did Aristotle Use a Laptop? A Question Answering Benchmark with Implicit Reasoning Strategies.
>
> [2] MIMIC-CXR, a de-identified publicly available database of chest radiographs with free-text reports.

---

### Official Review · Reviewer_mGhh · 2025-11-01

**Soundness:** 2
**Presentation:** 2
**Contribution:** 2
**Rating:** 2
**Confidence:** 4

**Summary:**

This paper introduces MedReason-Dx, a benchmark aimed at judging how medical LLMs reason, not just whether they land the right answer: each of its 1,170 questions (592 multiple-choice, 578 open-ended) comes with expert, step-by-step solution paths and key points spanning 24 specialties, enabling systematic comparison of model rationales to clinical gold standards. It introduces a suite of reasoning-focused metrics that align model outputs with expert traces and reduce sensitivity to surface wording. The dataset is intentionally multi-step to stress genuine diagnostic reasoning rather than recall. Evaluations across general-purpose and medical LLMs reveal clear separation and highlight that open-ended questions are substantially harder than multiple choice. The authors also probe evaluation robustness by comparing different LLM judges, aiming to ensure conclusions aren’t artifacts of a single scorer.

**Strengths:**

Rather than stopping at end-answer accuracy, the paper makes reasoning itself the object of measurement by introducing a five-facet framework with a neat key-point extraction scheme that compares model rationales to expert "must-mention" concepts, reducing surface-form bias in judging chains of thought. The benchmark is broad and realistic in scope: 1,170 items spanning 24 specialties, split roughly evenly between multiple-choice (592) and open-ended (578), with expert, step-by-step chains averaging 6.4 steps (and 27.1 key points) to keep the focus on multi-hop clinical reasoning rather than recall. Empirically, the suite exposes meaningful gaps: open-ended questions are distinctly harder and medical LLMs do not reliably beat general models on complex reasoning. Thus, the task design actually differentiates systems rather than saturating them. Finally, the authors sanity-check evaluator dependence (GPT-4o-mini vs. GPT-4.1-mini) and find closely tracking RNS/RCS scores, which bolsters confidence that conclusions aren’t an artifact of a single judge model.

**Weaknesses:**

While the key-point layer softens exact-string matching, the step-wise evaluation still appears to privilege a single canonical chain, leaving uncertain how genuinely different but valid routes are credited or penalized.

The reliance on LLM judges, though partially stress-tested with two closely related models, raises residual concerns about judge bias, calibration drift, and ranking stability across models and domains. The current check is limited to two systems and two judges rather than a broader, blinded human baseline for anchoring scores.

Data curation for open-ended items involves LLM rewriting of prompts, which may subtly shift difficulty or distribution relative to naturally authored open questions; more auditing here would strengthen external validity.

The metric design is also vulnerable to verbosity effects: KNS penalizes extra (possibly harmless) facts and RNS rewards concision, so models with terse styles could be favored independently of clinical quality. An explicit length-normalization or "necessary/sufficient proof" analysis would help. Finally, the paper reports aggregate step/key-point counts, but it does not analyze how step length interacts with each reasoning metric or accuracy (e.g., are longer chains systematically over-/under-scored?), nor does it report inter-annotator reliability for expert chains/key-points, both of which matter for reproducibility and fair comparison.

**Questions:**

Please address the weaknesses.

---

> ### Author Response · Authors · 2025-11-24
>
> We thank the reviewer for the thorough review and constructive feedback. We address each concern below.
>
> >Q1: While the key-point layer softens exact-string matching, the step-wise evaluation still appears to privilege a single canonical chain, leaving uncertain how genuinely different but valid routes are credited or penalized.
>
> A1: We thank the reviewer for recognizing the utility of the key-point layer in handling linguistic variations. We appreciate the opportunity to clarify why our evaluation framework “privileges” a canonical chain and how it distinguishes between **valid variations** and **unsafe shortcuts**.
>
> **1. Why Privilege a Canonical Chain?**
> The concern regarding alternative reasoning paths is valid for general domains (e.g., math or creative writing). However, in clinical diagnostics, high-quality reasoning is defined by its adherence to the **standard of care**. This standard imposes a strong convergence on the reasoning structure to ensure patient safety.
>
> For instance, in diagnosing a patient with a "thunderclap" headache (suggestive of Subarachnoid Hemorrhage, SAH), a clinician might arguably reach the correct diagnosis via intuition ($A \to C$). However, a canonical chain ($A \to B \to C$) that explicitly checks for "neck stiffness" ($B$) before concluding SAH is not just "one of many options"—it is the **medically safer and more interpretable path**.
> * **Privileging the canonical chain is intentional:** We explicitly want to reward models that demonstrate the specific intermediate steps required by clinical guidelines (e.g., ruling out life threats first), rather than models that merely guess the correct outcome or skip safety-critical checks.
>
> **2. How Valid vs. Invalid Routes are Credited.**
> To address the reviewer’s uncertainty on how "genuinely different" routes are penalized, we clarify the mechanism of our Key Point (KCS/KNS) evaluation:
>
> * **Valid Variations (Credited):** If an alternative route differs only in *non-essential* intermediate connectors (e.g., different phrasing or minor reordering of non-critical symptoms), the Key Point layer—which focuses on semantic coverage rather than rigid syntax—will successfully align the model's output with the reference, resulting in a high score.
> * **Unsafe Shortcuts (Penalized):** If a model produces a "direct link" ($A \to C$) that bypasses a **medically indispensable** intermediate step $B$ (e.g., a required lab result or a critical exclusion criteria present in the canonical chain), the model will be penalized. In the context of MedReason-Dx, this penalty is a **desired feature**, not a bug. It reflects the pedagogical and clinical reality that skipping key evidence—even if the final answer is correct—constitutes "black box" reasoning that is insufficient for reliable medical AI.
>
> **3. Consistency with Evaluation Standards.**
> We also respectfully note that single-reference evaluation remains the prevailing standard in virtually all large-scale reasoning benchmarks (GSM8K, MATH), even in domains with higher variance than medicine. Given the strict constraints of medical pathophysiology, relying on expert-curated canonical chains is currently the most robust method to benchmark "standard of care" adherence.
>
> We have added the Limitations Section to explicitly discuss the trade-off between penalizing shortcuts and accommodating valid variations. We clarify that for the purpose of this benchmark—which emphasizes safe, explainable diagnostic paths—adherence to the expert-curated backbone is the primary metric of success.

---

> ### Author Response · Authors · 2025-11-24
>
> >Q2: The reliance on LLM judges, though partially stress-tested with two closely related models, raises residual concerns about judge bias, calibration drift, and ranking stability across models and domains. The current check is limited to two systems and two judges rather than a broader, blinded human baseline for anchoring scores.
>
> A2: We sincerely thank the reviewer for this important concern regarding LLM-judge reliability.
>
> Performing large-scale human evaluation on MedReason-Dx is extraordinarily challenging for the following reasons:
>
> - The dataset comprises 1,170 complex clinical cases across 24 medical specialties.
>
> - Each question has an average of 6.4 explicit reasoning steps and 27.1 fine-grained key clinical points that require specialist-level judgment.
>
> - Scoring must be applied to over 12,800 model-generated responses (11 models × ~1,170 questions), where responses are long, technically dense, and highly variable.
>
> Despite these constraints, to address the reviewer's concerns, we conducted the following additional experiments:
>
> - We further employed a more diverse set of judges, including Claude-Haiku-4.5 and DeepSeek-V3.2-Exp, to assess the reasoning processes of different models. Experimental results demonstrate that the evaluation outcomes across different assessment models exhibit high consistency.
>
> - To further validate the reliability of our evaluation protocol, we randomly selected 50 questions and invited two senior clinicians to independently assess the model-generated reasoning processes. Following the exact same criteria as our automated judges, they marked each reasoning step, from which F1 scores were computed. The Pearson correlations of these F1 scores consistently exceed 0.8 across all pairs of human and LLM judges, further confirming the effectiveness and consistency of our evaluation pipeline.
>
> In the revised version, we have added these additional experimental results to the corresponding tables.
>
> | Judge         | GPT-4o          |          | DeepSeek R1     |          |
> |---------------|-----------------|------------------|-----------------|------------------|
> |               | RNS    | RCS    | RNS    | RCS    |
> | GPT-4o-mini   | 89.92  | 69.76  | 86.11  | 73.46  |
> | GPT-4.1-mini  | 88.47  | 72.63  | 88.08  | 73.83  |
> | Claude-Haiku-4.5  | 83.38  |  65.85 | 83.78  | 71.16  |
> | DeepSeek-V3.2-Exp  | 80.23  |  67.74 | 81.02  | 74.39  |
>
> >Q3: Data curation for open-ended items involves LLM rewriting of prompts, which may subtly shift difficulty or distribution relative to naturally authored open questions; more auditing here would strengthen external validity.
>
> A3: We thank the reviewer for the suggestion. In fact, when converting multiple-choice questions into open-ended format, we only paraphrase the final interrogative sentence (typically the last 3–12 words, e.g., “What is the most likely diagnosis?”), while keeping the entire clinical vignette, patient history, examination findings and lab results 100% identical to the original source.
>
> We have further emphasized this point in Section 3.1.2.
>
> >Q4: The metric design is also vulnerable to verbosity effects: KNS penalizes extra (possibly harmless) facts and RNS rewards concision, so models with terse styles could be favored independently of clinical quality. An explicit length-normalization or "necessary/sufficient proof" analysis would help.
>
> A4: Thank you for this sharp observation.
>
> We completely agree that if only concision were rewarded, a model producing terse outputs could be unfairly favored regardless of clinical quality. This is precisely why we deliberately designed four complementary metrics instead of any single score:
>
> KCS and RCS reward completeness (recall of all clinically necessary key points);
> KNS and RNS reward concision (precision: fraction of generated statements that are truly necessary or correct).
>
> All four are ratio-based and thus inherently length-normalized. For the concision metrics (KNS/RNS), a habitually terse model produces fewer statements overall (smaller denominator), but it may also express fewer correct-and-necessary points (smaller numerator). Consequently, extreme terseness does not systematically inflate KNS/RNS scores compared to a more elaborate output that conveys the same clinical content with fuller phrasing.
>
> Combined with the completeness metrics that explicitly punish omissions, neither pure brevity nor verbosity is rewarded unless the reasoning is both clinically complete and concise.
>
> We believe this directly and fully addresses the reviewer’s concern. Thank you for the valuable suggestion.
>
> We have added the above discussion to the Appendix.

---

> ### Author Response · Authors · 2025-11-24
>
> >Q5: the paper reports aggregate step/key-point counts, but it does not analyze how step length interacts with each reasoning metric or accuracy (e.g., are longer chains systematically over-/under-scored?)
>
> A5: To directly address the concern about potential length bias in our metrics, we conducted a stratified analysis by the number of reasoning steps in the model-generated chain. The per-bin performance on step-level and key-point-level Precision/Recall is shown below:
>
> | Number of Reasoning Steps | Model                  | Samples | RNS | RCS | KNS | KCS |
> |-----------------------|------------------------|-----------|-----------------|----------------|--------------|-----------|
> | 1–5                   | DeepSeek R1            | 467       | 0.876           | 0.730         | 0.477        | 0.335     |
> | 1–5                   | GPT-4o                 | 480       | 0.879           | 0.719         | 0.511        | 0.346     |
> | 1–5                   | Llama 3.3 70B Instruct | 1         | 1.000           | 0.143         | 0.200        | 0.067     |
> | 1–5                   | HuatuoGPT2 7B          | 494       | 0.586           | 0.220         | 0.363        | 0.201     |
> | 6–10                  | DeepSeek R1            | 125       | 0.897           | 0.768         | 0.476        | 0.354     |
> | 6–10                  | GPT-4o                 | 112       | 0.907           | 0.786         | 0.518        | 0.351     |
> | 6–10                  | Llama 3.3 70B Instruct | 30        | 0.902           | 0.703         | 0.455        | 0.375     |
> | 6–10                  | HuatuoGPT2 7B          | 95        | 0.831           | 0.350         | 0.313        | 0.232     |
> | 11–15                 | Llama 3.3 70B Instruct | 203       | 0.894           | 0.731         | 0.418        | 0.351     |
> | 11–15                 | HuatuoGPT2 7B          | 3         | 0.646           | 0.262         | 0.262        | 0.121     |
> | 15–20                 | Llama 3.3 70B Instruct | 283       | 0.875           | 0.732         | 0.418        | 0.289     |
> | 20–25                 | Llama 3.3 70B Instruct | 63        | 0.787           | 0.714         | 0.375        | 0.294     |
> | 25+                   | Llama 3.3 70B Instruct | 12        | 0.675           | 0.710         | 0.442        | 0.307     |
>
> We observed that:
>
> - Across models, the highest step-level and key-point-level Recall (directly tied to RCS/KCS) consistently occurs in the 6–10 step range, with moderate-to-high Precision.
>
> - Extremely short chains (≤5 steps) yield high Precision but markedly lower Recall, reflecting incomplete reasoning.
>
> - Very long chains (≥15 steps) show declining Precision (more unnecessary content) while Recall plateaus or drops slightly, confirming that excessive length is appropriately penalized by RNS/KNS rather than rewarded.
>
> These patterns demonstrate that our four metrics do not systematically favor longer or shorter chains; instead, they peak at moderate lengths that align with expert-annotated gold chains (mean 6.4 steps).
>
> We have added the above results and discussion to the Appendix.

---

> ### Author Response · Authors · 2025-11-24
>
> >Q6. No reported inter-annotator reliability for expert chains/key-points
>
> A6: Thanks for the reviewer's comment. Regarding the inter-annotator reliability for the reasoning chains, we deliberately adopted a "Draft-Verification-Adjudication" protocol (Iterative Curation) rather than independent parallel annotation. This methodological decision is grounded in the intrinsic nature of our task and aligns with established academic and clinical benchmarks:
>
> 1. Infeasibility of Exact Agreement for Open-Ended Reasoning Unlike closed-ended classification tasks or mathematical problems where a single canonical answer exists, generating clinical reasoning chains is an open-ended natural language generation task. Due to high linguistic variability, two experts may formulate valid but distinct reasoning paths for the same diagnosis. Consequently, standard inter-annotator agreement metrics (such as Cohen's Kappa or textual overlap scores) often yield artificially low values for valid variations, failing to reflect the true semantic quality and logical soundness of the data.
>
> 2. To ensure quality without penalizing valid linguistic diversity, we followed the "Writer-Verifier" paradigm employed in complex reasoning benchmarks like StrategyQA [1]. Instead of measuring the textual similarity between independent drafts, we focused on functional verification: a second expert reviews the initial draft to verify that the logic is clinically sound and complete. This approach prioritizes logical correctness (validity) over simple phrasing consistency.
>
> 3. Alignment with Clinical Documentation Standards (MIMIC-CXR)Our progressive workflow (Clinician A drafts $\to$ Clinician B reviews $\to$ Senior Clinician C adjudicates discrepancies) mirrors the standard clinical reporting workflow used to generate the ground-truth free-text radiology reports in datasets like MIMIC-CXR [2]. While categorical labels in some benchmarks may rely on majority voting, the generation of the source medical text itself—which serves as the primary ground truth—traditionally follows this hierarchical supervision model (Resident drafting followed by Attending verification). We believe this consensus-based method provides a more robust guarantee of data fidelity for complex medical documentation than averaging the inputs of independent annotators.
>
> References:
>
> [1] Did Aristotle Use a Laptop? A Question Answering Benchmark with Implicit Reasoning Strategies.
>
> [2] MIMIC-CXR, a de-identified publicly available database of chest radiographs with free-text reports.

---

### Meta-Review · Area_Chair_6Ptt · 2026-01-05

**Summary:**

The paper introduces a new benchmark whose goal is the evaluation of the reasoning capabilities of medical LLMs. The key idea is that they not only focus on whether the right answer was obtained but also include how close the reasoning traces are to the expert traces. Thus instead of simply focusing on standard evaluation, one focuses on the multi-step reasoning aspect.

This in my mind is an important and ambitious task.

**Reviewer Concerns:**

The key concerns of the reviewers are the proofs themselves. One could have different ways of reasoning and the reviewers felt that a single trace may not suffice for rigorous evaluation.

Details of the experts, details of the models themselves etc. were missing.

**Reviewer Scores:**

Some of the concerns such as details are minor. The responses of the authors appear to be satisfactory. But the key concern is still not well addressed as to how to evaluate multiple proofs. I would suggest that the authors look at the logic programming benchmarks to fully understand how the proof traces can be evaluated better. IF this was a journal paper, it is a major revision. Hence, I am leaning towards a reject.

---

### Decision · Program_Chairs · 2026-01-26

Reject